

# Development of a high-resolution emission inventory and its evaluation through air quality modeling for Jiangsu Province, China

Yaduan Zhou[1], Yu Zhao[1,2*], Pan Mao[1], Qiang Zhang[3], Jie Zhang[2,4], Liping Qiu[1], Yang Yang[1]

1. State Key Laboratory of Pollution Control & Resource Reuse and School of the Environment, Nanjing University, 163 Xianlin Ave., Nanjing, Jiangsu 210023, China

2. Jiangsu Collaborative Innovation Center of Atmospheric Environment and Equipment Technology (CICAEET), Nanjing University of Information Science & Technology, Jiangsu 210044, China

3. Ministry of Education Key Laboratory for Earth System Modeling, Center for Earth System Science, Tsinghua University, Beijing 100084, China

4. Jiangsu Provincial Academy of Environmental Science, 176 North Jiangdong Rd., Nanjing, Jiangsu 210036, China

*Corresponding author: Yu Zhao

Phone: 86-25-89680650; email: *yuzhao@nju.edu.cn*





1                             **ABSTRACT**

2       Improved emission inventories combining detailed source information are crucial for

better understanding the atmospheric chemistry and effectively making emission control
policies using air quality simulation, particularly at regional or local scales. With the
downscaled inventories directly applied, chemical transport model (CTM) might not be able
to well reproduce the evolution of atmospheric pollution process at small spatial scales. Using
the bottom-up approach, a high-resolution emission inventory was developed for Jiangsu
China, including $SO_2$, NOx, CO, $NH_3$, volatile organic compounds (VOCs), total suspended
particulates (TSP), $PM_{10}$, $PM_{2.5}$, black carbon (BC), organic carbon (OC), and $CO_2$. The key
parameters relevant to emission estimation for over 6000 industrial sources were investigated,
compiled and revised at plant level based on various data sources and on-site survey. As a
result, the emission fractions of point sources were significantly elevated for most species.
The improvement of this provincial inventory was evaluated through comparisons with other
inventories at larger spatial scales, using satellite observation and air quality modeling.
Compared to the downscaled Multi-resolution Emission Inventory for China (MEIC), the
spatial distribution of $NO_X$ emissions in our provincial inventory was more consistent with
summer tropospheric $NO_2$ VCDs observed from OMI, particularly for the grids with moderate
emission levels, implying the improved emission estimation for small and medium industrial
plants by this work. Three inventories (national, regional, and provincial by this work) were
applied in the Models-3/Community Multi-scale Air Quality (CMAQ) system for southern
Jiangsu, to evaluate the model performances with different emission inputs. The best
agreement between available ground observation and simulation was found when the
provincial inventory was applied, indicated by the smallest normalized mean bias (NMB) and
normalized mean errors (NME) for all the concerned species $SO_2$, $NO_2$, $O_3$ and $PM_{2.5}$. The
result thus implied the advantage of improved emission inventory at local scale for high
resolution air quality modeling. Under the unfavorable meteorology for pollution transport, in
particular, much higher $SO_2$ concentrations than observation were simulated for downtown
Nanjing (the capital city of Jiangsu) using the regional or national inventories, implying the
overestimation in urban emissions when the locations of large emitters were not fully



considered, and the densities of economy or population were simply applied to downscale or
allocate the emissions. With more accurate spatial distribution of emissions at city level, the
simulated concentrations using the provincial inventory were much closer to observation.
Sensitivity analysis of $PM_{2.5}$ and $O_3$ formation was conducted using the improved provincial
inventory through the Brute Force method. Iron & steel and cement plants were identified as
important contributors to the $PM_{2.5}$ concentrations in Nanjing (the capital city of Jiangsu). The
$O_3$ formation was VOCs-limited in southern Jiangsu, and the concentrations were negatively
correlated with $NO_X$ emissions in urban areas owing to the accumulated NOx from
transportation. More evaluations are further suggested for the impacts of speciation and
temporal and vertical distribution of emissions on air quality modeling at regional or local
scales in China.

**1 INTRODUCTION**

With rapid development of economy and growth of energy consumption, eastern China is

experiencing severe atmospheric pollution attributed to the large emissions of primary air
pollutants and the subsequent formation of secondary pollution, e.g., fine particles and $O_3$.
Relatively high concentrations of surface $PM_{2.5}$ were observed in eastern China based on the
national monitoring network (data source: http://106.37.208.233/), and only 9.5% out of 190
cities with the measurement data reported in 2014 met the National Ambient Air Quality
Standard (NAAQS), i.e., 35 μg/m$^3$ for annual $PM_{2.5}$ concentration (MEP, 2012). Under the
serious air pollution, series of measures have been conducted to reduce the pollutant
emissions and to improve the air quality across the country. Issued in 2013, for example, the
National Air Pollution Prevention Action Plan required strict emission controls on both
industry and transportation sectors, and aimed to achieve a 25%, 20% and 15% reduction of
annual $PM_{2.5}$ concentration for Beijing-Tianjin-Hebei (JJJ), Yangtze River Delta (YRD), and
Pearl River Delta (PRD) region from 2012 to 2017, respectively. Given the non-linear
response of ambient concentrations to emissions, chemical transport modeling (CTM) has
been widely applied to study the mechanisms of complex pollution processes and the impacts
of emission abatement (Zhang et al., 2006; Streets et al., 2007; B. Zhao et al., 2013; Zhang et



al., 2012). As the key input of CTM, therefore, improved emission inventories, particularly at
regional or local scales, become important for scientific air quality simulation and effective
policy making.
Progress has been increasingly achieved in emission inventory studies for China.
Compared to earlier national emission inventories including those for Transport and Chemical
Evolution over the Pacific Mission (TRACE-P, Streets et al., 2003), Intercontinental Chemical
Transport Experiment-Phase B (INTEX-B, Zhang et al., 2009), and Regional Emission
inventory in Asia (REAS, Ohara et al., 2007; Kurokawa et al., 2013), Tsinghua University
developed     the     Multi-resolution     Emission     Inventory     for     China     (MEIC,
http://www.meicmodel.org/), in which the information of large power plants and cement
factories was investigated and the uncertainties of emission estimation for those typical
sources were reduced (Wang et al., 2014). Besides, high-resolution emission inventories at
regional and city scales were gradually established in the developed regions JJJ, YRD and
PRD, attributed to better data support and stronger need to combat air pollution (Zheng et al.,
2009; S. Wang et al., 2010; Huang et al., 2011; B. Zhao et al., 2012; Zhao et al., 2015).
Resulting from various methods and data sources, clear discrepancies exist in different
emission inventories in China, both at national (Y. Zhao et al., 2013; Xia et al., 2016) and
local scales (Zhao et al., 2015). When applied in CTM, the uncertainties in emission
estimation would inevitably lead to bias in air quality simulation, besides the errors of
meteorological field modeling and deficiencies of built-in atmospheric chemical mechanisms
(Zheng et al., 2012). Based on the Models-3/Community Multi-scale Air Quality (CMAQ)
system, for example, Zhang et al. (2014) simulated $PM_{2.5}$ and $O_3$ concentrations in
southeastern United States using the different versions of national emission inventory (NEI),
and compared the results with several ground observational datasets. The model performance
with updated inventory (NEI05) was much better than that with old one (NEI99), indicating
the impacts of emission inventory on the accuracy of CMAQ simulations. Han et al. (2015)
conducted $NO_2$ vertical column simulation for China with CMAQ, and found that the
modeled results using INTEX-B inventory were closer to satellite observation than those
using REAS. At regional or local scales, emission inventory that incorporates the detailed
information of individual sources is assumed to have advantages in air quality research prior





to downscaled national inventory that generally applied regional average levels of emission
factors due to unavailability of data (Zhao et al., 2015). The benefits of improved emission
estimation, spatial and temporal distribution, or chemical speciation of pollutants, however,
have not been sufficiently confirmed with CTM. Recently, Yin et al. (2015) conducted CMAQ
simulation on $O_3$ using updated VOC emission inventory for PRD, implying that the reduced
uncertainties of total emission level and spatial distribution could improve the model
performance compared with ground observation.

We select Jiangsu, a typical province with well-developed industry in eastern China, to

develop and evaluate the high-resolution emission inventory. The geographic location and
cities of the province are illustrated in Figure S1 in the supplement. With a total area of 107
200 $km^2$ and population of 79.2 million in 2012, Jiangsu was the first ranked province in gross
domestic product (GDP) per capita in China (NBSC, 2013a; JSNBS, 2013). It accounted for
8.0%, 7.6%, 8.9%, and 10.2% of the country's power generation, cement, pig iron, and crude
steel production in 2012, respectively (NBSC, 2013b). Intensive energy consumption and
industry resulted in heavy air pollution: all the 13 cities had their annual average
concentrations of $PM_{2.5}$ exceeding the NAAQS in 2012, with the highest reaching 74 $\mu g/m^3$ in
the capital city, Nanjing. Clear uncertainties exist in current multi-scale emission inventories.
Zhao et al. (2015), for example, estimated Nanjing's $SO_2$ and $PM_{2.5}$ emissions at 165 and 71
Gg in 2012, respectively, while the results by Fu et al. (2013) were 131.8 and 35.3 Gg,
implying the necessity of improvement and assessment of regional emission inventory, for
both scientific and policy implication. In this work, a comprehensive emission inventory for
Jiangsu with high temporal and spatial resolutions was first established with the best available
data of local emission sources incorporated. This provincial emission inventory was then
compared with other inventories and satellite observation to test its improvement on emission
estimation and spatial distribution. CMAQ was further applied to indicate the advantage of
the provincial inventory prior to downscaled national and regional ones. In particular, the
impacts of spatial distribution of emissions on model performance were analyzed for period
with unfavorable meteorological condition. Finally, the improved inventory was applied for
sensitivity analysis on regional $PM_{2.5}$ and $O_3$ formation.



# 2 DATA AND METHODS


**2.1 Methodology of provincial emission inventory development**
The emissions of gaseous pollutants ($SO_2$, NOx, CO, $NH_3$ and VOCs), greenhouse gas
$CO_2$, particulate matter (total suspended particulates (TSP), $PM_{10}$ and $PM_{2.5}$) and its chemical
compositions (black carbon, BC and organic carbon, OC) of anthropogenic origin in Jiangsu
were estimated with a bottom-up method. Emission sources were classified into seven main
categories, including power plants, industry, solvent usage, transportation, residential &
commercial, agriculture and others. Industry was subdivided into iron & steel, cement, and
other industry including nonferrous metal smelting, brick and lime kilns, chemical industry
and other industry boilers. Residential & commercial sector included household combustion
of fossil fuel and biofuel. Agriculture included livestock and fertilizer usage. Open biomass
burning, cooking, and waste (water) disposal, were considered as other sources. The detailed
categories were summarized in Table S1 in the supplement. For each category, point, mobile
and area sources were defined depending on the detailed levels of information and the
emission characteristics, For point sources, information on emission factor and activity level
was investigated and compiled for individual plants, and the annual emissions of atmospheric
pollutants were calculated using Eq. (1), as described in Zhao et al (2015) :
$$E_i = \sum_{j,m} AL_{i,j,m} \times EF_{i,j,m} \times (1 - \eta_{i,j,m}) \qquad (1)$$
where $i$, $j$ and $m$ represented the pollutant species, individual plant, and fuel/technology type,
respectively; $AL$ was the activity level data; $EF$ was the uncontrolled emission factor; and $\eta$
was removal efficiency of air pollutant control device.
Regarded as mobile sources, the emissions of on-road transportation were calculated by
the CORPERT model (EEA, 2012) and then spatially allocated based on the road net
information of the province. Area sources include non-road transportation, solvent use,
residential & commercial sector, agriculture, and small industry plants without detailed
information collected. The emissions from non-road transportation and agriculture were
estimated following the methods by Zhang et al. (2010) and Dong et al. (2009), respectively.
**2.2 Activity level**
Most of coals in Jiangsu were used by power and industry sectors, and household





accounted for only 0.3% of total coal consumption in the province in 2012 (JSNBS, 2013),
indicating the significance to reduce the uncertainties of emission estimation for power and
industry plants. Therefore a comprehensive database for power and industrial sectors was
established with the information collected and modified from the official environmental
statistics, Pollution Source Census (PSC, internal data), and on-site survey on large emitters.
Parameters including geographical location, combustion/production technology,
fuel/burner/boiler type, installed air pollution control device (APCD) and its removal
efficiency were investigated for individual plants. Totally 6750 plants were identified as point
sources, including 191 power plants, 185 iron & steel plants, 231 cement factories, 707 lime
and brick factories, 365 chemical plants and 5071 other industrial factories, as illustrated in
Figure S2 in the supplement. In particular, the kilns for combustion and factories for
calcination were separately investigated for cement production, and 25% of cement plants
contained the both processes. For power, cement, and iron & steel sectors, the aggregated
activity levels compiled plant by plant, i.e., the coal consumption of power generation, and
the production of cement, clinker, coke, pig iron, and crude steel, were estimated at 108%,
95%, 120%, 109%, 104%, and 98% of the provincial statistics, respectively (JSNBS, 2013).
The comparison indicates, on one hand, that larger activity levels would be obtained based on
detailed investigation of individual emission sources than official statistics for power and
most processes of iron & steel sectors. On the other hand, almost completed investigation on
point sources was conducted for those sectors, as very small fractions of activities (5% for
cement and 2% for steel production) had to be estimated as area sources. For other industrial
sectors, smaller fractions of point sources were obtained, e.g., 32% and 36% for ammonia and
sulfuric acid production, respectively.

For on-road transportation, the input parameters of COPERT 4 include regional

meteorological information, vehicle population by type, fleet composition by control stage
(China I–IV, equivalent to Euro I–IV), average vehicle speeds, and annual average kilometers
traveled (VKT). Monthly mean temperature and relative humidity were obtained from the
China Meteorological Data Sharing Service System (http://www.escience.gov.cn).
Populations of different vehicle types were derived from statistical yearbooks by city and then
adjusted in accordance with the model requirement. The fleet composition by control stage





was obtained from the survey by local government (internal data, Zhao et al., 2015). Vehicle
speed by road type (i.e., freeway, arterial and residential road) and VKT by vehicle type were
determined according to previous studies (Cai and Xie, 2007; Wang et al., 2008) and the
guidebook of emission inventory development for Chinese cities (He, 2015). For area sources,
the coal consumption of residential & commercial activities was directly taken from National
Energy Statistic Yearbook (NBSC, 2013c), while that of small industrial plants were
calculated as the coal consumption of total industry from the provincial energy balance
(NBSC, 2013c) minus the coal consumption of industrial point sources. The original data on
the activity levels of agriculture, solvent use, non-road transportation and open biomass
burning were obtained from the provincial or city statistical yearbooks (JSBNS, 2013).
**2.3 Emission factor**
Following previous studies (Zhao et al., 2008; 2010; 2011; 2012a; 2012b; Y. Zhao et al.,
2013), an emission factor database for Jiangsu was established with detailed information and
available results of emission measurements on local sources incorporated. For power sector,
parameters relevant to emission factors were obtained at individual plant level including
installed capacity, fuel type and quality (e.g., sulfur and ash content), combustion technology,
and the type and removal efficiencies of APCDs. In particular, the information of APCD
installation obtained from provincial environmental statistics and on-site survey was further
corrected according to the official documents on APCD projects of power plants published by
Ministry        of        Environment        Protection        of        China
(http://www.zhb.gov.cn/gkml/hbb/bgg/201305/t20130506_251654.htm). As summarized in
Table S2 in the supplement, the application rates of flue gas desulfurization (FGD), selective
catalytic reduction (SCR)/selective non-catalytic reduction (SNCR), and dust collectors for
Jiangsu's power plants in 2012 were 97%, 57% and 99% in terms of coal consumption, and
the average removal efficiencies of $SO_2$, NOx and TSP weighted by coal consumption were
calculated at 83.3%, 37.1% and 98.0%, respectively. Combining all the above-mentioned
information, the emission factors for individual plant and facility were calculated using the
methods developed by Zhao et al. (2010).
For iron & steel production, emission factors of the four main manufacturing processes





(coking, sintering, pig iron production, and steel making) were estimated combining the
unabated emission factors from previous database (Zhao et al., 2011; Y. Zhao et al., 2013) and
the investigated information on penetrations and removal efficiencies of APCDs at plant level.
Provided in Table S2, 64.3% of Jiangsu's iron & steel plants installed FGD in 2012 and the
average $SO_2$ removal efficiency was estimated at 78.0%. Dust collectors were installed at
almost all the furnaces for pig iron production and steel making, with the averages of PM
removal efficiency estimated at 96% and 94%, respectively. For cement production, emission
factors were calculated for the two main processes, coal combustion and calcination,
following Lei et al. (2011). With dust collectors installed at 99% of plants, the average of
overall removal efficiency on TSP was estimated at 97.3% according to our plant-by-plant
investigation (Table S2).
For area sources, emission factors for non-road transportation were obtained from Zhang
et al. (2010), Ye et al. (2014) and Fu et al. (2012). Emission factors for household fossil fuel
and biofuel combustion were from the summary of field measurements in Y. Zhao et al.
(2013). For agricultural activities including livestock and fertilizer use, emission factors were
obtained from Dong et al. (2009) and Yin et al. (2010). Emission factors of VOCs were
mainly from Wei et al. (2009) with update for typical sources based on limited local
measurements and survey (Bo et al., 2008; EEA, 2013; Xia et al., 2014).
**2.4 Temporal and spatial distributions**
The monthly variations of emissions from power plants and industrial sources were
assumed to be dominated by with the variations of electricity generation and typical industrial
production, respectively, and those data were obtained from National Bureau of Statistics of
China (http://data.stats.gov.cn/). As the real-time monitoring on urban traffic was unavailable
for the whole province, the temporal distribution of emissions from on-road vehicles in other
cities was considered to the same as Nanjing (Zhao et al., 2015). For other sources, the
temporal distributions in Shanghai investigated by Li et al. (2011).
Different parameters were used to conduct the spatial allocation of emissions by sector.
Latitude and longitude of each point source collected from PSC were checked and revised
according to Google Earth to avoid the unexpected errors in the existing database. The



densities of GDP and population were applied to allocate the emissions from industrial area
sources, and residential and commercial sources, respectively. Emissions from on-road
transportation were allocated based on the road net by city. As the ship flow was unavailable,
the widths of Yangtze River and the Grand Cannel within Jiangsu were used as indicators for
ship emissions. Emissions from open biomass burning were allocated by the locations and
brightness of agricultural fire spots observed by MODIS (Moderate Resolution Imaging
Spectroradiometer, https://earthdata.nasa.gov/data/near-real-time-data/firms). $NH_3$ emissions
from livestock and fertilizer use were allocated by the density of rural population.
**2.5 Configuration of air quality modeling**
The Models-3/Community Multi-scale Air Quality (CMAQ) version 4.7.1 was applied to
evaluate the emission inventory for Jiangsu. As shown in Figure 1, three one-way nested
domain modeling was conducted, and the spatial resolutions were set at 27, 9 and 3 km
respectively in Lambert Conformal Conic projection, centered at (110° E, 34° N) with two
true latitudes 25°N and 40°N. The mother domain (D1, 180×130 cells) covered most part of
China, Japan and the whole Korea and part of other country. Jiangsu, Zhejiang, Shanghai,
Anhui and parts of other provinces were at the second modeling region (D2, 118×97 cells).
The third (D3, 124×70 cells) covered the mega city Shanghai and six most developed cities
in southern Jiangsu including Nanjing, Changzhou, Zhenjiang, Wuxi, Suzhou and Nantong.
The simulation period was selected from October 1 to 31, 2012, with the first five days
chosen as spin-up period to provide initial conditions for later simulations.
Meteorological fields were provided by the Weather Research and Forecasting Model
(WRF) version 3.4 with the main physical options set as Fu et al. (2014), and the outputs were
transferred by meteorology chemistry interface professor (MCIP) version 4.2 into the
chemistry transport module in CMAQ (CCTM). In WRF, the U.S. Geological Survey (USGS)
database was adapted as terrain and land use data, and the first guess field of meteorological
modeling was provided by the final analysis dataset (ds083.2) from National Centers for
Environmental Prediction (NCEP). Statistical indicators including Bias, Index of Agreement
(IOA), and root mean squared error (RMSE) were applied to evaluate the performance of
WRF modeling against observation (Baker, 2004; Zhang et al., 2006). Ground observations in



three hours interval at meteorological stations were downloaded from National Climatic Data
Center (NCDC), including 43 stations in the second modeling domain D2 and 7 stations in the
innermost domain D3 (as labeled in Figure 1). The statistics of those indicators for wind
speed and direction at 10 m (WS10 and WD10), temperature at 2 m (T2) and relative
humidity at 2 m (RH2) for October 2012 in D2 and D3 were summarized in Table S3.
Discrepancies between WRF simulations and ground observations were within acceptable
range (Emory et al., 2001) and comparable to the results of other studies (Wang et al., 2014).
Better agreements were found for simulations of T2 and RH2 than WS10 and WD10. In spite
of moderate overestimation by 0.3% and 3.5% in T2 and RH2, the IOA of those two variables
were 0.97 and 0.90, indicating the high consistency with observationa. Slightly higher than
observation in D2 and D3, simulated WS10 might enhance the diffusion process of pollutants
in atmosphere eventually and thus lead to underestimation in pollutant concentrations. For
WD10, the bias between simulations and observations was 3.6 degree in D3 within the
benchmark range (Emory et al., 2001).
The carbon bond gas-phase mechanism (CB05) and AERO5 aerosol module were
adopted in all the CMAQ modules. The initial and boundary conditions for first modeling
domain was the default clean profile, while for nested domain they were extracted from the
CCTM outputs of its mother domain. Anthropogenic emissions used for domains D1 and D2
were obtained from the downscaled MEIC with an original spatial resolution of 0.25°×0.25°.
For Jiangsu domain in D3, three inventories, i.e., downscaled MEIC, the regional inventory of
YRD by Fu et al. (2013), and the provincial inventory developed in this work, were used to
test the modeling performance and potential improvement in emission estimation. In addition,
biogenic emission inventory were from the Model Emissions of Gases and Aerosols from
Nature developed under the Monitoring Atmospheric Composition and Climate project
(MEGAN-MACC, Sindelarova et al., 2014), and the emission inventories of Cl, HCl and
lightning $NO_X$ were from the Global Emissions Initiative (GEIA, Price et al., 1997). The
vertical distribution of emissions was determined by source following L. Wang et al. (2010).



**3 RESULTS**
**3.1 Emission estimation and sector contribution**
The total annual emissions of $SO_2$, NOx, CO, TSP, $PM_{10}$, $PM_{2.5}$, BC, OC, $CO_2$, $NH_3$ and
VOCs were calculated at 1142, 1642, 7680, 2606, 1394, 941, 57, 138, 860458, 1100 and 1747
Gg for Jiangsu in 2012, respectively. The emissions by city were provided in Table 1. In
general, higher emissions were found in cities in southern Jiangsu with large population and
intensive economy and industry than those in northern Jiangsu. Taking 52% of the provincial
industrial GDP, Suzhou, Nanjing, and Wuxi were estimated to collectively account for 41%,
41%, 35%, 31%, 43% and 39% of the total emissions of $SO_2$, NOx, CO, $PM_{2.5}$, $CO_2$ and
VOCs, respectively. Xuzhou, different from other cities in northern Jiangsu, had a relative
high emissions of pollutants due to its well development of large-scale industry. Because of
the active agricultural development, $NH_3$ emissions in Huai'an and Nantong were estimated at
195.9 and 187.1 Gg, significantly higher than other cities.
Shown in Figure 2 is the detailed sector contribution of pollutants from point, mobile
(on-road transportation) and area sources. The point sources including power and industrial
plants contributed 84%, 71%, 55%, 83%, 75%, 64%, 41%, 31%, 83%, 2% and 36%, to the
total emissions of $SO_2$, NOx, CO, TSP, $PM_{10}$, $PM_{2.5}$, BC, OC, $CO_2$, $NH_3$ and VOCs,
respectively. The emission fractions of point sources were notably larger than those in other
regional inventories (Fu, 2009; Tang et al., 2012; B. Zhao et al., 2012), resulting mainly from
the compiling and application of information on individual power and industrial plants from
varied data sources. Defined as area source, open biomass burning contributed 12%, 19%,
23%, 11% and 41% to the total CO, $PM_{10}$, $PM_{2.5}$, BC and OC, respectively.
The dominant contributors to $SO_2$ were power plant, iron & steel and other industry, with
the emission fractions estimated at 38%, 10% and 45%, respectively. Although the coal
consumption in power sector was 3.5 times larger than that in other industry sector (cement
and iron & steel production excluded, JSNBS, 2013), smaller contribution to $SO_2$ emissions
were found for coal-fired power plants, implying the benefits of strict control on $SO_2$
emissions from power sector. As shown in Table S2, the application rate and average $SO_2$
removal efficiency of FGD in power sector were significantly higher than those in other



industry, suggesting the improvement in $SO_2$ abatement for industrial coal combustion other
than power plants would be an effective measure to further reduce the emissions.
Power sector was the largest source for $NO_X$, contributing 41% to the total emissions,
while the share of coal consumption of the sector reached 65% (JSNBS, 2013). It thus implied
the tightened controls from implementation of new emission standard (GB13223-2011) and
improved use of SCR/SNCR on power plants since 2011 compared to other sectors. Compiled
from unit level, the average NOx removal efficiency of SCR/SNCR was calculated at 37% for
Jiangsu's power plants in 2012 (Table S2), while Tian et al. (2013) estimated the values for
SCR and SNCR at 70% and 25%, respectively, indicating the differences in assessment of
emission controls for power sector between the provincial and national emission inventories
with varied data sources. Transportation (including on-road and non-road) was estimated to be
the second largest sector for $NO_X$ emissions, with the share to the total emissions calculated at
24%. Without specific control measures, cement and other industry were estimated to account
for 7% and 17% of total NOx emissions.
CO was mainly generated from the manufacturing processes in iron & steel plants. The
production of pig iron and crude steel in Jiangsu accounted for 9% and 10% to the national
total in 2012, respectively (NBS, 2013), and was higher than other provinces in China except
Hebei. Due to the intensive iron & steel industry, the contribution of the sector to the
provincial total CO emissions was estimated at 35%. Residential biofuel combustion, open
biomass burning and on-road transportation were also large contributors to CO with the
emission fractions calculated as 24%, 12% and 11% respectively.
For particles, iron & steel and cement production were estimated to be the largest sources,
contributing 24% and 27% to the total emissions of $PM_{10}$, and 27% and 19% to $PM_{2.5}$,
respectively. Even with the largest coal consumption among all the sectors, the emissions
from power plants were relatively small (6% and 4% to $PM_{10}$ and $PM_{2.5}$ emissions,
respectively), resulting mainly from the relatively high penetrations and removal efficiencies
of dust collectors. Great differences existed in the sector distribution of BC and OC emissions.
Iron & steel was estimated to be the largest source of BC, while open biomass burning and
biofuel burning in residential stoves dominated OC, with the shares estimated at 41% and
29%, respectively. Moreover, as BC exhausted from the diesel engines was demonstrated to





be higher than OC in previous situ measurements (He et al., 2015), BC emissions from non-road transportation (agricultural machines, rural vehicles, ships and construction machines) was estimated more than twice larger than OC.

For VOCs, solvent use and other industry including oil refinery, chemical industry and combustion were identified as the largest sources contributing 30% and 29% to total emissions, respectively. In particular, oil refinery and chemical engineering collectively accounted for 74% of the emissions of other industry. Due to lack of investigation on chemical industry plants, the fraction of area sources to the emissions of other industry reached 35%. Transportation and residential cooking are estimated to contribute 12% and 4% to total VOCs emissions, respectively. Livestock and fertilizer use were the two dominating sources of $NH_3$, with the shares to total emissions estimated at 47% and 45%, respectively. For industry, ammonia production was the main source accounting for half of $NH_3$ emissions.

The spatial distribution of $SO_2$, NOx, CO, $PM_{2.5}$, VOCs and $NH_3$ emissions at a resolution of 3×3km were illustrated in Figure S3 in the supplement. Outstandingly high emissions of $SO_2$, NOx, $PM_{2.5}$ and VOC indicated the existence of large industrial plants, particularly in Suzhou, Nanjing and Wuxi along with the Yangtze River. For CO and NOx, large emissions were distributed along the road net in the province, reflecting the important contribution of on-road transportation. Unlike other pollutants, high $NH_3$ emissions were more evenly distributed in rural areas as dominated by agricultural activities.

**3.2 Comparisons with other studies**

Figure 3 compares the emission estimations for Jiangsu between our provincial inventory and previous studies including two regional inventories (Fu et al., 2013; Li et al., 2011) and two national ones (MEIC; Xia et al., 2016). Note this work and Xia et al. (2016) reported the numbers for 2012, while Fu et al. (2013), Li et al. (2011) and MEIC for 2010. As the emissions from open biomass burning were not included in other inventories except Fu et al. (2013) and this work, two values labeled as A and B were provided for our provincial inventory indicating the emissions without and with biomass open burning, respectively. While provincial economy and energy data were generally applied in all the national/regional inventories, information of individual large emitters were incorporated as well in MEIC, Fu et



381 al. (2013) and Li et al. (2011). For example, the emissions of big plants for power generation,

382 iron & steel and cement production in Jiangsu were partially investigated in Fu et al. (2013)

383 and Li et al. (2011). For MEIC, large fraction of emissions from power generation sector was

384 calculated plant by plant with relatively good data availability, while emissions from other

385 industrial sectors were basically calculated at regional average and spatially allocated as area

386 sources. The results in Fu et al. (2013) were generally smaller than those in other two

387 inventories for 2010.

388   Attributed mainly to the improved use of FGD, the total $SO_2$ emissions were estimated to

389 decline from 2010 to 2012 for the whole country (Xia et al., 2016) and typical city in Jiangsu

390 (Zhao et al., 2015). It was reasonable to some extent that the $SO_2$ emissions in Jiangsu

391 estimated in this work for 2012 was less than the 2010 results by Li et al. (2011) and MEIC.

392 Our estimation was 15% lower than the result for Jiangsu extracted from the national

393 inventory by Xia et al. (2016), due mainly to the discrepancies in the penetration and $SO_2$

394 removal efficiency of FGD applied in the two inventories. Such information was obtained at

395 provincial or national average level by Xia et al. (2016), in contrast to the provincial

396 inventory based on investigation at plant level. For example, Xia et al. (2016) assumed that

397 the penetration rates of wet and dry FGD technologies in coal-fired power sector were 83%

398 and 5% in 2012, with the removal efficiencies estimated at 80% and 40%, respectively, and

399 that there was not any $SO_2$ control in the remaining 11% of installed capacity at all. According

400 to our plant-based investigation, the controls in Jiangsu were clearly enhanced, as shown in

401 Table S2. As a result, $SO_2$ emissions from power sector was calculated at 430.0 Gg for

402 Jiangsu 2012 in this work, 42% lower than those in Xia et al. (2016). The result for 2012 in

403 our provincial inventory, however, is very close to the estimation by MEIC for 2010 (437.4

404 Gg), even though the coal consumption of power generation increased 29% for the period

405 2010-2012 (JSNBS, 2013). Besides the uncertainty in emission estimation from varied data

406 sources of the two inventories, the improved use of FGD in the sector could be an important

407 reason for the restrained emissions. Similar fact was found for Nanjing, the capital city of

408 Jiangsu, that the $SO_2$ emissions of power generation calculated at city level kept stable along

409 with a 25% growth of coal consumption from 2010 to 2012 (Zhao et al., 2015).

410   NOx emissions in our provincial inventory was slightly higher than those of Li et al.



(2011) and clearly lower than the two national inventories. The major difference between the
provincial inventory and MEIC was from industry, attributed probably to the application of
varied emission factors. With different methods and data sources for certain sectors, the NOx
emissions from industry were calculated at 388.1 and 566.2 Gg respectively by this work and
Xia et al. (2016). For on-road transportation, the emission factors were estimated using
CORPERT in this work, while they were obtained from limited domestic measurements in
Xia et al. (2016). That was also the most important reason for the discrepancies in CO
emission estimation between the two studies. For 2010, the $NO_X$ emissions estimated by Fu et
al. (2013) was 18% and 36% lower than those by Li et al. (2011) and MEIC, resulting mainly
from the higher application rate and removal efficiency of SCR/SNCR technologies for power
sector used in Fu et al. (2013).

The $PM_{2.5}$ and $PM_{10}$ emissions in the provincial inventory were estimated to be 6% and

23% higher than those of Xia et al. (2016), and the sector contributions were notably different
in the two inventories. For example, industry was estimated to contribute 77% and 80% of
$PM_{2.5}$ and $PM_{10}$ in the provincial inventory, much larger than the fractions at 45% and 52% by
Xia et al. (2016), respectively. In this work, the $PM_{2.5}$ and $PM_{10}$ emissions from cement
production were calculated at 181 and 384 Gg, i.e., 2.5 and 2.0 times to those in Xia et al.
(2016), and the analogue numbers for iron & steel production were 134 and 263 Gg, and 1.8
and 1.7 times, respectively. The discrepancies resulted mainly from the inconsistent
penetration rates and removal efficiencies of dust collectors determined at national level and
from on-site survey at provincial level. Taking cement as an example, all the plants were
assumed to be installed with dust collectors, and the national average removal efficiency at
99.3% was applied in Xia et al. (2016), clearly larger than that from plant-by-plant survey as
shown in Table S2. Note that the particle emissions in the provincial inventory were estimated
higher than those in national ones including MEIC and Xia et al. (2016), while the gaseous
pollutant emissions were lower except for $NH_3$ and $CO_2$. It thus implied that the control of
$SO_2$ and NOx in Jiangsu were stronger than the national average level but weaker for particles.
Compared to the emissions for 2010 estimated by other studies, the $PM_{2.5}$ and $PM_{10}$ in our
provincial inventory were 58% and 56% larger than Fu et al. (2013) (biomass open burning
included), and 24% and 25% larger than Li et al. (2011) (biomass open burning excluded),



respectively, beyond the growth rate of 20% for coal consumption during 2010-2012 (NBS,

2011; 2013).

The $NH_3$ emissions of Fu et al. (2013) and Li et al. (2011) were close to each other, while

MEIC was only half of them for 2010. Using the results for 2006 from Huang et al. (2012),
MEIC made a very low estimation in $NH_3$ emissions from livestock. The $NH_3$ emissions for
2012 in this work was calculated 11% and 22% larger than the results for 2010 by Fu et al.
(2013) and Li et al. (2011), respectively. The agricultural GDP increased 35% from 2010 to
2012 in Jiangsu (JSNBS, 2013), thus the growth of activity levels were expected to result in
enhanced emissions, as very little progress was achieved for $NH_3$ control for these years.
**3.3 Analysis of spatial distribution of emissions from given sectors**

To further explore the discrepancies in emission estimation and spatial distribution from

varied data and emission allocation methods, comparisons between MEIC and our provincial
inventory were conducted for pollutants from typical sources, including $SO_2$ from power
plants, $NO_X$ from transportation, and $PM_{2.5}$ from industry. The estimates in this work were
reallocated into the 0.25°×0.25° grids, consistent with the spatial resolution of MEIC, and the
correlation coefficients for emissions in all the grids can be calculated, as shown in Figure 4.
Due mainly to the relatively transparent and easily available information of power plants,
good consistency was found for $SO_2$ emissions from power sector in the two inventories, with
the correlation coefficient calculated at 0.7 (Figure 4a). Even though the fundamental
information of power plants in China is more accessible than other industry sources,
mismatches still exist in different data sources. For example, some emission hotspots in our
provincial inventory were not totally identical with those in MEIC in Suzhou, Nantong and
Nanjing. In contrast to plant-by-plant investigation, the data from existing statistics at national
level could not fully track the actual changes in the emitters, e.g., operation of new-built units,
shutting down the small ones, or relocation of individual plants. In MEIC, moreover, the $SO_2$
emissions in several grids were estimated extremely small (less than 1 Mg), indicating that
part of emissions from power sector was still allocated as area sources based on density of
GDP or population. In contrast, all the plants were identified as point sources in the provincial
inventory, based on the thorough investigation on individual sources.



For NO$_X$ from transportation, the correlation coefficient was calculated at 0.8, indicating
an even better consistency than SO$_2$ from power plants between the two inventories (Figure
4b). Although the difference in total emissions was small between our provincial inventory
(682 Gg) and MEIC (722 Gg), the estimation of MEIC was notably higher than our result for
northern Jiangsu including Yancheng, Huai'an and Suqian, implying the impacts from
different ways for emission allocation. In this work, emissions from on-road vehicles were
calculated and allocated based on road net that incorporates the information of transportation
flow by road grade for each city. For non-road sources, large fraction of emissions was
allocated based on the GDP density incorporated with land-use type. In national emission
inventories, however, the emissions were first calculated at provincial level, and then
downscaled at certain horizontal resolution. Despite of the discrepancies, it could be indicated
by the relatively high spatial correlation between the two inventories that using GDP as proxy
for emission allocation would be acceptable when detailed information on road net and
transportation flow was unavailable, since vehicles were largely concentrated in downtown
with the intensive economic activity.
For PM$_{2.5}$ from industry, the correlation coefficient was calculated at 0.335, significantly
lower than those mentioned above, indicating larger discrepancy in spatial distribution of
industrial emissions between provincial and national inventories compared to power and
transportation sectors. As shown in Figure 4c, the emission hotspots in the provincial
inventory are highly consistent with the locations of large industrial PM$_{2.5}$ emitters (more than
10 Gg) estimated in this work, while the emission in MEIC were more distributed in
developed cities (e.g., Suzhou) with high density of population or economy. Along with fast
urbanization, super industrial sources have gradually been relocated to the rural or suburban
areas, and the spatial correlation between industrial emissions and population could thus be
weakened. In our provincial inventory, most industrial enterprises were identified as point
sources, with the key parameters including geographic location, activity level and removal
efficiency of dust collector investigated and corrected at plant level. In MEIC, the emissions
were calculated using parameters at regional average level and allocated as area sources
according to densities of population and/or economic activity. Without detailed information
for individual sources, it might lead to errors in emission estimation and spatial distribution at





regional or local scale. According to the survey at plant level, for example, only 20% of the lime factories were installed dust collectors in Jiangsu 2012, much lower than the value (roughly 90%) assumed in national inventories. As a result, the $PM_{2.5}$ emissions from industry were calculated at 570 Gg in our provincial inventory, 78% higher than those of MEIC.

## 4 ASSESSMENT OF THE PROVINCIAL EMISSION INVENTORY

### 4.1 Evaluation of spatial distribution of $NO_X$ emissions with satellite observation

Troposphere $NO_2$ vertical column density (VCD) retrieved from Ozone Monitoring Instrument (OMI) by the Royal Netherlands Meteorological Institute (Boersma et al., 2011) was employed to test the spatial distribution of NOx emissions in MEIC and this work. $NO_2$ VCDs in summer were used due to the short lifetime of $NO_2$ in atmosphere at high temperature and the difficulty in accumulation for primary emissions with strong air convection. In addition, the summer prevailing wind for Jiangsu was generally from southeast where Shanghai and Zhejiang Province are located (see Figure S1 for the locations of the three regions). Total $NO_X$ emissions of Jiangsu were estimated to be 65% and 282% larger than those of Shanghai and Zhejiang Shanghai in MEIC, and local sources were expected to dominate the pollution for the province (Cheng et al., 2011). As Mijling et al. (2013) illustrated satellite observations could be used to evaluate the primary emissions for regions where $NO_2$ VCDs were mainly affected by local emissions, it was thus feasible to apply the OMI $NO_2$ VCDs in Jiangsu to assess its NOx emissions.

$NO_2$ VCDs in July 2010 and 2012 with original spatial resolution of 0.125°×0.125° were used for comparisons with the emissions in MEIC and our provincial inventory, respectively. To be consistent with MEIC, the emissions in our provincial inventory and the $NO_2$ VCDs from OMI were first upscaled to 0.25°×0.25° for the purpose of visualization and correlation analysis. As can be seen in Figure 5a and 5b, clear reduction in summer $NO_2$ VCDs was found in southern Jiangsu from 2010 to 2012, indicating the benefits of efforts on $NO_X$ abatement since 2011. The $NO_2$ VCDs in the area along the Yangtze River were notably higher than that in other regions, attributed possibly to the substantial emissions from vessels and small captive power plants of the chemical and refinery industrial parks along the river





without stringent controls as big power plants. Shown in Figure 5c and 5d are the spatial
distributions of Jiangsu's $NO_X$ emissions in MEIC and our provincial inventory, respectively,
and the emission hotspots were generally consistent between the two inventories. Figure 5e
and 5f shows the linear regression results between $NO_2$ VCDs and NOx emissions in MEIC
and the provincial inventory, respectively. The correlation coefficients between VCDs and
emissions were separately provided for all the grids and grids in different emission intervals,
i.e., top 50%, 50%-75%, and last 25%.

The correlation coefficient between $NO_2$ VCDs and NOx emissions from the provincial

inventory was 0.534, close to that between $NO_2$ VCDs and MEIC at 0.531. The result
indicated that there was no significant difference in spatial distribution of emissions between
the national and provincial inventories at the relatively low horizontal resolution. However,
great discrepancies existed when the correlation analysis was conducted for grids in different
emission intervals. As shown in Figure 5e, the correlation coefficients between VCDs and
MEIC emissions were calculated at 0.24 and 0.34, respectively, for the top 50% (20 grids with
emissions ranged 32-121 Gg) and last 25% of gridded emissions (161 grids with emissions
ranged 0-12 Gg) . For our provincial emission inventory, the correlation coefficients were
estimated slightly higher at 0.28 and 0.38, respectively, for the top 50% (18 grids with
emissions ranged from 30-75 Gg) and last 25% of gridded emissions (176 grids with
emissions ranged 0-12 Gg). Moreover, the coefficient between $NO_2$ VCDs and gridded
emissions for the 50% -75% interval in provincial inventory was 0.26, while negative value
(-0.07) was calculated for MEIC, indicating that the emission estimation for areas with small
and medium plants in the provincial inventory was more consistent with satellite observation.
To better quantify the emissions at local scale, the results revealed the practical significance of
careful investigation on individual small industrial plants that were usually identified as area
sources due to lack of detailed information in national or regional inventories.

**4.2 Evaluation of multi-scale inventories with CMAQ**

As mentioned in Section 2.5, anthropogenic emission inventories at provincial, regional

and national scales were applied respectively to explore the impacts of emission input on the
performance of city-scale air quality simulation using CMAQ. With the original horizontal



resolutions at $0.25^{o}\times0.25^{o}$ and $4\times4$ km, respectively, national (MEIC) and YRD regional
inventories (Fu et al., 2013) were reallocated into the D3 of CCTM modeling at $3\times3$ km
(Figure 1), consistent with our provincial inventory. The vertical and temporal distributions of
the two inventories were assumed to be same as those of our provincial inventory, as indicated
in Section 2.4. Given the very limited data accessible on air quality for the province in 2012,
the available observation data at nine state-operated monitoring sites in Nanjing including six
urban sites (Xuanwumen (XWM), Shanxilu (SXL), Zhonghuamen (ZHM), Ruijinlu (RJL),
Caochangmen (CCM), and Maigaoqiao (MGQ)) and three suburban sites (Pukou (PKS),
Xianlin (XLS) and Olympic sports center (OSC)) were applied to evaluate the simulation
performances with different emission inputs (see locations of the nine sites in Figure 1).
The hourly ground concentrations from observation and CMAQ simulation for October
2012, expressed as the averages for all the monitoring sites in Nanjing, were compared and
illustrated in Figure S4 in the supplement for $SO_2$, $NO_2$, $O_3$ and $PM_{2.5}$. Even though all the
simulations could well reproduce the time variation of each species, discrepancies existed
when different anthropogenic emission inventories were used. The simulated $SO_2$ and $NO_2$
concentrations using the provincial inventory were notably lower than those with other two
inventories. In addition, simulations could hardly catch the high $PM_{2.5}$ and $O_3$ concentrations
in heavy polluted episode. For example, the average $PM_{2.5}$ ground concentration during
October $21^{st}$-$23^{rd}$ and $28^{th}$-$29^{th}$ were simulated at 40 and 31 $\mu g/m^3$, 1.4 and 3.2 times lower
than observations. Two statistical indicators, normalized mean bias (NMB) and normalized
mean error (NME), were applied to evaluate the model performance (Zhang et al., 2006), as
summarized in Table 2. Among all the species, the best simulation performance was found for
$NO_2$, with the NMBs ranged within $\pm30\%$ for different emission. In general, simulations
using the provincial emission inventory performed notably better than those with national and
regional ones for all the species, and the NMEs and NMBs were calculated at 47%, 33%,
44%%, 52% and -10%, -14%, -25%, -43% for $SO_2$, $NO_2$, $O_3$, $PM_{2.5}$ respectively, comparable
to previous U.S. studies (Zhang et al., 2006; Wang et al., 2009). The result thus partly
confirmed that air quality simulation at local or regional scale would be largely improved
when detailed information on individual sources could be incorporated in the emission
inventory. Compared to primary pollutants $SO_2$ and $NO_2$, however, species with strong





secondary formation process ($PM_{2.5}$ and $O_3$ in this case) were clearly under predicted by
CMAQ, no matter which inventory was applied. Lack of dust emissions in inventories might
be one reason for underestimation of $PM_{2.5}$. Moreover, as the significant composition of $PM_{2.5}$
in eastern China (Yang et al., 2011), secondary organic and inorganic aerosols might be under
predicted attribute to the weakness of chemical mechanisms in the version of CMAQ
including the transformation of sulfate and the formation of secondary organic aerosols (Wang
et al., 2009). For $O_3$ simulation, better performance was found at suburban sites than urban
sites, and the lower simulated concentrations than observation could possibly come from the
underestimation in precursor VOCs emissions. For example, the NMB was estimated at -26%
for PKS, where many chemical industrial plants were located nearby. In addition, the
uncertainty of $NO_X$ emission estimation might also contribute to the discrepancy. As indicated
by the data from available continuous emission monitoring systems on Jiangsu's power plants,
the $NO_X$ emission factors of power sector applied in current inventory might be overestimated
for 2012 (unpublished).

The total emissions of $SO_2$ and NOx in Jiangsu estimated by Fu et al. (2013) was 1126

and 1257 Gg, i.e., 9% and 22% lower than the results of our provincial inventory, respectively.
Using the regional inventory by Fu et al. (2013), much higher concentrations of $SO_2$ and $NO_2$
were simulated than observation at the monitoring sites, with the NMBs calculated at 74%
and 30%, respectively. Even with larger emissions, in contrast, the NMBs for simulation with
our provincial inventory were -10% and -14%, indicating lower simulated concentrations than
observation. This result implies the possible impacts of spatial distributions of emissions on
air quality modeling. In regional inventory, densities of population and economic activities
were generally applied to allocate large fraction of emissions, leading to particularly high
emissions in urban areas, as the economy and population was generally centralized in
downtown. Given all the monitoring sites in Nanjing are located in urban or suburban areas,
air quality simulation using regional emission inventory was thus liable to over predict the
ground concentrations at those sites.

Spatial distributions of the monthly mean for simulated concentrations using national,

regional and provincial inventories were plotted for $SO_2$, $NO_2$, $PM_{2.5}$ and $O_3$ in Figure 6, and
the differences between simulations with varied emissions were shown in Figure 7. As the



MEIC emissions were greatly averaged when they were directly downscaled from
0.25°×0.25° to 3×3 km, the simulated high concentrations using MEIC were broadly
distributed in the modeling domain and commonly located in downtown (as indicated in
Figure S1), with the large emitters hardly identified (Figure 6a). For results using the regional
and provincial inventories, there were several grids with notably outstanding simulated
concentrations indicating the existence of large emitters (Figure 6b and 6c), and differences
with the simulation using MEIC were induced (Figure 7a and 7b). As shown in Figure 7c,
moreover, clear differences were also detected between simulations using regional and
provincial inventories, implying the discrepancy in allocations of high emissions between the
two inventories. With the locations of large power, iron & steel, and cement plants
incorporated, the YRD regional emission inventory by Fu et al. (2013) allocated a large
fraction of emissions from industries as area sources. In contrast, the emissions from most
power and industrial plants were calculated based on source-specific information and were
precisely allocated in the provincial inventory, avoiding particularly the emission
overestimation in downtown. In addition, the simulated $NO_2$ and $O_3$ concentrations for
regions outside Jiangsu (i.e., Shanghai and part of Zhejiang and Anhui) using the provincial
inventory were 22% lower and 40% higher in average than those using the regional one,
respectively (Figure 7c), although same emissions (Fu et al., 2013) were used outside Jiangsu
for the two inventories. The result indicated that both local and regional emissions were
important for the simulations of the secondary pollutant like $O_3$. Total VOCs emissions for
Jiangsu were estimated at 1740 Gg in MEIC, slight higher than those in the regional (1659 Gg)
and provincial inventory (1617 Gg), while the simulated monthly mean $O_3$ concentrations
within Jiangsu using MEIC were notably lower than those using the latter two emissions.
Categorized by CB05, differences in chemical compositions of VOCs could be found in the
three inventories. For example, the emissions of ethene (ETH) and ethanol (ETHA) with
relatively high ozone formation potential in the provincial inventory were 44% and 209%
higher than those in MEIC, respectively. Therefore, the total emission amount, spatial
distribution of emissions, and the chemical compositions of precursors are all crucial to the
accuracy of ozone simulations, and further analysis on those factors are suggested.





### 4.3 Improved SO$_2$ simulation under special meteorological condition


To further examine the simulated concentration response to varied emission inputs at
local scale, the simulated SO$_2$ concentrations using national, regional and provincial
inventories were compared with observation at three monitoring sites in downtown Nanjing
(XWH, RJL and ZHM) for 6$^{th}$ -14$^{th}$ October 2012, as illustrated in Figure 8. The simulated
concentrations using our provincial inventory were the most consistent with observation,
while apparent overestimation was found for the simulations using national or regional
inventories. At 8 pm October 9 (local time), in particular, the SO$_2$ concentrations were
observed at 33, 12, and 14 μg/m$^3$ at XWH, RJL and ZHM sites, respectively, while the
simulated concentrations were respectively simulated at 205, 246 and 228 μg/m$^3$ using MEIC,
i.e., 5-19 times higher than the observation. The analogue numbers with regional inventory by
Fu et al. (2013) even reached 550, 477 and 476μg/m$^3$, i.e., 15-38 times higher than
observation. Although concentrations remained over predicted, better performance was
achieved when the improved provincial inventory was used, implying its advantage prior to
national or regional ones in the high-resolution air quality modeling. The discrepancies in
emissions and the simulated meteorological condition including wind velocity and height of
planetary boundary layer (PBL) were inspected to understand the very high concentrations
from simulation.
Figure S5 in the supplement shows the simulated wind fileds from 2 pm on 9$^{th}$ to 5am on
10$^{th}$ October. From 2 pm on 9$^{th}$ October 9, WS10 in downtown Nanjing started to decline
gradually and reached the minimum of 0.22 m/s at 8 pm , simply not beneficial for the
horizontal convection of atmosphere. In addition, the monthly average of PBL height at XWH
was simulated at 485 m at day and 140 m at night in October. From 5pm on 9$^{th}$ to 10am on
10$^{th}$, however, the average PBL height decreased to 39m, with the minimum simulated at 32
m at 11pm on 9$^{th}$ , seriously restricting the vertical diffusion of pollutants. Under the
meteorological condition that horizontal and vertical movement of atmosphere were limited,
primary pollutants from large emitters would be easily accumulated over time, possibly
leading to high concentrations for areas close to the emission sources. In this case, therefore,
the simulated SO$_2$ concentrations would be largely influenced by the emissions from local and
nearby sources, as discussed below.



The total SO$_2$ emissions in Nanjing were estimated at 141 Gg in the provincial inventory,
2% and 7% higher than those of national and regional ones respectively. Without big
difference in total amount, large discrepancies in spatial distribution existed in those
inventories, particularly at high horizontal resolution as shown in Figure 9. Downscaled from
0.25°×0.25° to 3×3 km, grids with similar emissions were clustered for MEIC and spatial
variations in emissions could hardly be detected other than the hotspot in downtown (Figure
9c). Notably lower emissions in downtown Nanjing were found in our provincial inventory
than the regional one (Figure 9a and 9b). In addition, the grid with maximum SO$_2$ emissions
(15.7 Gg) in the provincial inventory was in the northwestern of Nanjing where a super power
plant was located, labeled as the black star (point A) in Figure 9. As a comparison, the grid
with the maximum SO$_2$ emissions in the regional inventory labeled as the black triangle (point
B) in Figure 9 was adjacent to the location of A, and its emissions were calculated to be only
28% of the result in the provincial one. Given no other super emitters located nearby, we
expected that the discrepancy resulted mainly from the varied emission estimation and
positioning for the same power plant in the two inventories. According to on-site survey, only
one unit out of two for the plant was installed with FGD, and the SO$_2$ emissions of the plant
was estimated at 13.6 Gg, accounting for 87% of the total emissions in the grid. In contrast, a
higher FGD installation rate at 85% was uniformly assumed for the power sector in the
regional inventory by Fu et al. (2013), leading to possible underestimation in emissions for
the plant. The comparison implied that detailed information compiled from individual plants
was crucial for estimation and spatial distribution of pollutant emissions at local scales. SO$_2$
emissions at given monitoring sites were extracted from the gridded national, regional and
provincial inventories and summarized in Table 3. As most large SO$_2$ emitters were located in
suburban or rural areas, relatively small emissions were found in the provincial inventory for
downtown Nanjing where the monitoring sites were located. As large fractions of emissions
were allocated by the density of economy and population, however, the SO$_2$ emissions in the
regional emission inventory were estimated at 1791, 1721, 1918, and 1635 Mg at XWH, RJL,
ZHM and CCM sites, which were 4-5 times higher than those of our provincial inventory. In
MEIC, the emissions at XWH, RJL, ZHM and MGQ sites were identically estimated at 1298
Mg from the downscaling approach, and they were also much larger than those in the





provincial inventory. Given the unfavorable condition of pollutant transport for 9th-10th
October, the overestimation in local emissions around the downtown monitoring sites in the
national and regional inventories thus lead to terribly high simulated concentrations, while the
results using the provincial one were much more reasonable. The comparison confirmed the
benefits of precise quantification of emissions on local air quality modeling.
Despite of significant improvement, overestimation in $SO_2$ concentrations still existed in
the simulation with our provincial inventory, attributed possibly to the error of meteorology
modeling. Here we selected XWH site as an example to conduct the back trajectory analysis
using HYSPLIT model (http://ready.arl.noaa.gov/HYSPLIT.php). Shown in Figure S6 in the
supplement, the air mass reaching the site at 50 m altitude came mainly from northeast at
11pm on 9th October. However, it was inconsistent with WRF modeling results, which
indicated the dominating wind was from northwest (150°-170°) at that time. As mentioned
above, a big power plant was located northwest to XWH (Figure 9a), and the site might partly
be influenced by the large emissions from the plant and enhanced concentrations would then
be obtained when northwest wind was simulated.

**5 SENSITIVITY ANALYSIS OF PM$_{2.5}$ AND OZONE FORMATION IN NANJING**
Using the improved provincial inventory, the sensitivity of PM$_{2.5}$ and O$_3$ concentrations
to emissions were further analyzed through the Brute-Force method (BFM, Dunker et al.,
1996). For PM$_{2.5}$, four simulation scenarios were designed: Scenario B (the base case) in
which the emissions from all types of sources are included; and Scenarios S1, S2, and S3 in
which the pollutant emissions of power, iron & steel and cement plants in D3 were zero out,
respectively. The changes in simulated PM$_{2.5}$ ground concentrations in S1, S2, and S3
compared to those in base case for October 2012 are illustrated in Figure S7 in the supplement.
The average concentration increments in urban area of Nanjing caused by power, iron &steel,
and cement plants were calculated respectively at 3, 11 and 7 μg/m$^3$, accounting for 6%, 26%
and 16% of the monthly mean PM$_{2.5}$ concentrations, and the maximum increments within the
domain reached 10, 72, and 25 μg/m$^3$, respectively. Given the tiny emission fraction of power
sector for primary PM$_{2.5}$ (4% in Jiangsu Province) and the small share in the ground layers



(15% for $1^{st}$ plus $2^{nd}$ vertical layers), its contribution to $PM_{2.5}$ ground concentration was
notably lower than those of iron & steel and cement. Summarized in Table 4 are the
contributions of power, iron & steel, cement sectors to monthly mean $PM_{2.5}$ at the nine
monitoring sites in Nanjing, October 2012. The contributions of the three sectors to average
$PM_{2.5}$ concentrations at all the sites were estimated at 8%, 13% and 9%, respectively. Since all
the sites are located in the urban or suburban areas, the estimated $PM_{2.5}$ contributions at
individual site varied slightly to each other. Besides monthly mean, the hourly maximum and
minimum contributions are provided as well in Table 4. The largest hourly contributions from
power, iron & steel and cement plants to $PM_{2.5}$ concentrations were 65% at PKS, 89% at
MGQ and 58% at both CCM and OCS, respectively. The contributions became negative at 2
pm on $26^{th}$ October with average $PM_{2.5}$ concentration of all the sites observed as 164 μg/m$^3$
and simulated as 151 μg/m$^3$ under the base case, i.e., increased particle concentrations were
simulated at the moment when emissions from given sector was turned off. The result
indicated, on one hand, the relatively high uncertainty of simulation for heavy PM pollution
episode dominated by regional transport. On the other hand, as the simulated increments were
mostly from the elevated sulfate ($SO_4^{2-}$), nitrate ($NO_3^-$) and ammonium ($NH_4^+$), the negative
contributions might also be caused by the complex chemical mechanisms of $SO_2$ and NOx
reactions with $NH_3$ under the $NH_3$-rich condition in YRD (Wang et al., 2011). Intensive
real-time observation on chemical composition of $PM_{2.5}$ is thus recommended to better
capture and analyze the process.

To explore the sensitivity of $O_3$ formation to its precursor emissions, two scenarios were

set besides the base case: the VOC-abatement scenario with 50% reduction of all
anthropogenic VOCs emissions in D3 (Scenario P1), and the NOx-abatement scenario with
50% reduction of NOx in D3 (Scenario P2). Shown in Figure S8 in the supplement were the
average $O_3$ concentration changes from October $6^{th}$ to October $15^{th}$. The simulated $O_3$ average
concentration from 11am to 5pm declined significantly under Scenario P1, with the maximum
reduction at 54 μg/m$^3$ (Figure S8a) within D3, and changes in the downwind region were
greater than the upwind. In contrast, the concentrations were generally enhanced under P2
with the maximum increment at 19 μg/m$^3$. Similar variation pattern was found for 1-hour
maximum $O_3$ concentration in Figure S8b and monthly mean concentration in Figure S8c.





The 1-hour maximum $O_3$ concentrations in most downwind area of Shanghai and southern
Jiangsu decreased 10-20 μg/m$^3$ with the reduction in VOCs emissions, and the concentrations
would generally increase 10-30 μg/m$^3$ with the $NO_X$ reduction. The similar patterns of $O_3$
concentration variation in urban and downwind areas in D3 under P1 or P2 scenario indicated
that the $O_3$ formation was VOCs-limited in all those areas in southern Jiangsu. Therefore,
VOC emission abatement could be effective for $O_3$ pollution control in southern Jiangsu,
while $NO_X$ abatement might aggravate the pollution in autumn.
The temporal changes in the simulated $O_3$ concentrations between the P1/P2 and base
scenarios at urban (XWH, SXL, RJL, MGQ, ZHM and CCM) and suburban sites (XLS, OCS
and PKS) in Nanjing were illustrated for October 6$^{th}$-16$^{th}$ in Figure 10. Simulated $O_3$
concentrations at urban and suburban sites were generally decreased once the VOC emissions
declined and the maximum hourly reduction reached 77.3 and 49.6 μg/m$^3$, respectively. In
contrast, concentrations were elevated with the NOx emission reduction and the maximum
growth were 78.7 and 15.4 μg/m$^3$, respectively. Under VOCs-limited regime, in general, the
$O_3$ concentration would be little sensitive to the changes in NOx unless it was rich enough to
turn to the negative correlation with $O_3$. Therefore, due to the intensive NOx emissions from
on-road transportation in downtown Nanjing, the variations of $O_3$ concentrations in P2
scenario at urban monitoring sites were notably greater than those at suburban sites.

**6 CONCLUSIONS**

The bottom-up approach was applied to develop a high-resolution emission inventory for
Jiangsu, with substantial detailed information on local sources incorporated. Key parameters
relevant to emission estimation were examined and revised plant by plant including
geographic position, energy consumption and removal efficiencies of APCDs from various
data sources and on-site survey on large emitters. Compared to previous studies, the emission
fractions of point sources were significantly enhanced, except for $NH_3$ and OC, which are
mainly from agriculture activities and biomass open burning, respectively. As lower removal
efficiencies of dust collectors were obtained from local investigation, larger primary PM
emissions were estimated in our provincial inventory than other national or regional ones.



Moreover, clear discrepancy existed in spatial distribution of industrial $PM_{2.5}$ emissions
between this work and the national inventory MEIC, indicating the uncertainty of emission
downscaling from coarse horizontal resolution. The spatial distribution of $NO_X$ emissions in
the provincial inventory was more consistent with summer tropospheric $NO_2$ VCDs observed
from OMI than that of MEIC, particularly for the emissions from small and medium industrial
plants. WRF-CMAQ air quality modeling system was set up to evaluate the reliability and
improvement of the provincial emission inventory by comparing the simulation performance
with that using a national (MEIC) and regional one. Among the three inventories, the best
agreement was found between the observation and simulation with the provincial one for all
the concerned species at the nine monitoring sites in Nanjing, while underestimation existed
particularly for $PM_{2.5}$ and $O_3$ that were strongly influenced by secondary formation. Under the
unfavorable meteorology of pollutant transport, extremely high $SO_2$ concentrations were
simulated using the regional and national inventories, while the results using provincial one
were much closer to the observation. The results indicated the advantage of improved
estimation and spatial distribution of emissions on air quality modeling at regional or local
scales. The improved provincial inventory was further applied for the sensitivity analysis on
$PM_{2.5}$ and $O_3$ formation using BFM simulation, and provided the preliminary results for the
policy making of regional haze and photochemical pollution control in southern Jiangsu.

Limitations remained in the current inventory. Attributed to unavailability of detailed

information, the weekly and hourly variations of emissions could not be fully tracked for each
city, and the vertical distribution of emissions by sector, depending mainly on the stack height,
temperature and flow of flue gas, could not be accurately determined. Instead, empirical data
from previous work (Li et al., 2011; L. Wang et al., 2010; 2014) had to be applied, which
might be inconsistent with the reality. In addition, some sources were not included in the
current inventory, e.g., fugitive dust emissions from construction sites and road transportation,
resulting from lack of reliable data and thereby potentially large uncertainties in the emission
estimation at provincial level. Finally, the effects of source profiles on air quality modeling,
e.g., the speciation of primary $PM_{2.5}$ and VOC, were not evaluated. As they are important on
the formation of $O_3$ and secondary particles, more investigations on typical sources and
evaluation through chemistry transport modeling are suggested in the future.




## ACKNOWLEDGEMENT


This work was sponsored by the Natural Science Foundation of China (41575142),
Natural Science Foundation of Jiangsu (BK20140020), Jiangsu Science and Technology
Support Program (SBE2014070918), and Special Research Program of Environmental
Protection for Commonweal (201509004). We would like to acknowledge Litao Wang from
Hebei University of Engineering, Jia Xing from Tsinghua University for the assistance in
CMAQ model, and Xiao Fu from Tsinghua University for providing the emission inventory
for Yangtze River Delta region, China.

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





**FIGURE CAPTIONS**
**Figure 1. Modeling domain and locations of 43 meteorological and 9 air quality**
**monitoring sites.**
**Figure 2. Source contributions to total estimated emissions by species in Jiangsu 2012.**
**Colors indicate the sectors and the shade patterns indicate the source type (point, mobile**
**and area).**
**Figure 3. Comparison between the emissions estimated in this work and other studies**
**for Jiangsu. A and B indicate the emissions without and with open biomass burning,**
**respectively.**
**Figure 4. Spatial distributions (a-c) and linear regression (d) of certain pollutant**
**emissions from typical sources estimated in our provincial inventory and MEIC. (a) SO$_2$**
**from power plant; (b) NOx from transportation; (c) PM$_{2.5}$ from industry. The black**
**points indicate the locations of plants with PM$_{2.5}$ emissions larger than 10 Gg estimated**
**in this work.**
**Figure 5. Spatial distributions of NO$_2$ VCDs observed by OMI in Jiangsu in 2010 (a) and**
**2012 (b), and those of Jiangsu's NOx emissions from MEIC (c) and our provincial**
**inventory (d) at the resolution of 0.25$^\circ$×0.25$^\circ$. Linear regressions of gridded VCDs and**
**emissions are illustrated for MEIC (e) and our provincial inventory (f).**
**Figure 6. Spatial distributions of the monthly means of simulated SO$_2$, NO$_2$, PM$_{2.5}$ and**
**O$_3$ concentrations using the national, regional and provincial emission inventories for**
**October 2012.**
**Figure 7. The differences in the monthly means of simulated SO$_2$, NO$_2$, PM$_{2.5}$ and O$_3$**
**concentrations using different emission inventories: (a) Provincial–national; (b)**
**Regional–national; and (c) provincial–regional. The black star A and triangle B referred**
**to the locations of grids with maximum SO$_2$ emissions in provincial and regional**
**inventory.**
**Figure 8. The observed and simulated hourly SO$_2$ concentrations at 3-hour interval**
**using the national, regional, and provincial inventories at XWH (a), RJL (b), and ZHM**
**(c) from October 6[th] to 13[th], 2012.**
**Figure 9. Spatial distributions of the estimated SO$_2$ emissions in Nanjing at the**
**resolution of 3×3km in the provincial (a), regional (b) and national emission inventory**
**(c). The black dots indicate the locations of given air quality monitoring sites. The black**
**star (point A) indicates the location of the power plant with the largest SO$_2$ emissions**
**estimated in the provincial inventory. The black triangle (point B) indicates the**
**speculated position of the same power plant in the regional inventory.**
**Figure 10. The changes in simulated O$_3$ concentrations at urban (XWH, SXL, RJL,**
**MGQ, ZHM, and CCM) and suburban air quality monitoring sites (XLS, OSC, and**
**PKS) in Nanjing under P1 (a) and P2 (b) scenarios compared to the base case for 6[th]-16[th]**
**October 2012.**



**TABLES**

**Table 1. The estimated annual emissions by city for Jiangsu 2012 (unit: million metric tons (Tg) for CO₂ and kilo metric tons (Gg) for other species).**

| City | SO₂ | NOx | CO | TSP | PM₁₀ | PM₂.₅ | BC | OC | CO₂ | NH₃ | VOCs |
|---|---|---|---|---|---|---|---|---|---|---|---|
| Southern | | | | | | | | | | | |
| Nanjing | 140.6 | 210.5 | 742.9 | 157.3 | 97.3 | 75.8 | 5.8 | 7.1 | 97.1 | 64.2 | 221.9 |
| Suzhou | 220.8 | 286.7 | 1383.3 | 380.6 | 194.9 | 137.3 | 9.8 | 11.0 | 184.4 | 144.8 | 297.8 |
| Wuxi | 107.7 | 180.0 | 545.5 | 271.3 | 126.9 | 77.2 | 3.4 | 9.6 | 84.5 | 24.2 | 167.2 |
| Changzhou | 104.0 | 107.7 | 734.6 | 413.3 | 194.6 | 126.2 | 3.6 | 7.5 | 65.2 | 33.4 | 104.2 |
| Zhenjiang | 44.0 | 89.6 | 231.6 | 143.3 | 66.8 | 40.9 | 1.9 | 6.6 | 53.0 | 38.1 | 55.4 |
| Central | | | | | | | | | | | |
| Nantong | 76.8 | 130.1 | 443.4 | 244.9 | 108.2 | 66.0 | 4.8 | 9.3 | 51.6 | 181.7 | 162.2 |
| Yangzhou | 55.3 | 93.9 | 310.7 | 54.1 | 39.9 | 31.1 | 2.6 | 8.5 | 52.1 | 83.1 | 82.5 |
| Taizhou | 56.6 | 70.5 | 315.1 | 207.6 | 98.2 | 52.3 | 2.7 | 8.8 | 31.4 | 100.7 | 76.9 |
| Northen | | | | | | | | | | | |
| Xuzhou | 138.9 | 232.5 | 805.5 | 223.2 | 146.5 | 101.9 | 6.1 | 19.1 | 139.2 | 49.2 | 161.2 |
| Huai'an | 52.2 | 61.5 | 590.0 | 97.4 | 64.5 | 49.5 | 3.7 | 12.0 | 32.5 | 195.9 | 78.6 |
| Yancheng | 49.9 | 78.5 | 639.7 | 203.8 | 111.8 | 72.0 | 5.6 | 16.1 | 28.2 | 101.0 | 185.0 |
| Lianyungang | 60.6 | 61.0 | 571.1 | 131.0 | 89.0 | 68.6 | 3.9 | 11.9 | 28.3 | 25.1 | 78.0 |
| Suqian | 34.1 | 39.7 | 366.8 | 77.8 | 55.5 | 42.3 | 3.2 | 11.0 | 12.9 | 59.1 | 76.4 |
| | | | | | | | | | | | |
| Total | 1141.5 | 1642.2 | 7680.0 | 2605.6 | 1394.0 | 941.1 | 57.0 | 138.5 | 860.5 | 1100.3 | 1747.3 |





**Table 2. Model performance statistics for concentrations of given species from observation and CMAQ simulation using the national, regional and provincial inventories at the nine air quality monitoring sites in Nanjing for October 2012.**

| Pollutants | National (MEIC) | | Regional (Fu et al., 2013) | | Provincial (this work) | |
|---|---|---|---|---|---|---|
| | NMB | NME | NMB | NME | NMB | NME |
| SO$_2$ | 48.45% | 76.53% | 74.08% | 95.04% | -9.97% | 47.49% |
| NO$_2$ | 21.02% | 35.99% | 29.84% | 43.45% | -14.47% | 33.22% |
| O$_3$ | -65.55% | 68.57% | -53.93% | 61.59% | -24.98% | 44.29% |
| PM$_{2.5}$ | -51.63% | 55.32% | -49.16% | 56.00% | -43.64% | 51.81% |

Note: NMB and NME were calculated using following equations ($P$ and $O$ indicate the results from modeling prediction and observation, respectively):

$$NMB = \frac{\sum_{i=1}^{n}(P_i - O_i)}{\sum_{i=1}^{n} O_i} \times 100\% \text{ ; } NME = \frac{\sum_{i=1}^{n}|P_i - O_i|}{\sum_{i=1}^{n} O_i} \times 100\%$$





**Table 3. The annual SO$_2$ emissions estimated in three inventories at given air quality monitoring sites in downtown Nanjing.**

| SO$_2$/Mg | National (MEIC) | Regional (Fu et al., 2013) | Provincial (this work) |
|---|---|---|---|
| XWH | 1297.5 | 1790.9 | 411.0 |
| RJL | 1297.5 | 1720.8 | 303.1 |
| ZHM | 1297.5 | 1918.3 | 396.2 |
| CCM | 928.6 | 1635.3 | 371.8 |
| MGQ | 1297.5 | 478.6 | 395.0 |



**Table 4. The monthly mean contributions of power, iron & steel and cement plants to the PM$_{2.5}$ concentrations at the air quality monitoring sites in Nanjing in October 2012.**

| Monitoring site | Contri. of power (%) | | | Contri. of iron & steel (%) | | | Contri. of cement (%) | | |
|---|---|---|---|---|---|---|---|---|---|
| | Max. | Min. | Ave. | Max. | Min. | Ave. | Max. | Min. | Ave. |
| XWH/SXL | 52 | -6 | 8 | 82 | -2 | 14 | 43 | -1 | 8 |
| RJL | 42 | -6 | 7 | 79 | 0 | 11 | 44 | 0 | 9 |
| ZHM | 44 | -5 | 7 | 71 | -3 | 12 | 48 | 0 | 9 |
| CCM | 32 | -8 | 7 | 83 | -4 | 13 | 58 | -5 | 8 |
| MGQ | 58 | -5 | 9 | 89 | -2 | 8 | 35 | -5 | 7 |
| XLS | 35 | -5 | 7 | 67 | -3 | 10 | 57 | 0 | 10 |
| PKS | 65 | -6 | 7 | 77 | -1 | 11 | 45 | -1 | 7 |
| OCS | 33 | -7 | 7 | 87 | 0 | 12 | 58 | 0 | 8 |

Note: Max., min., ave. and contri. indicate maximum, minimum, average and contribution, respectively.





**Figure 1**

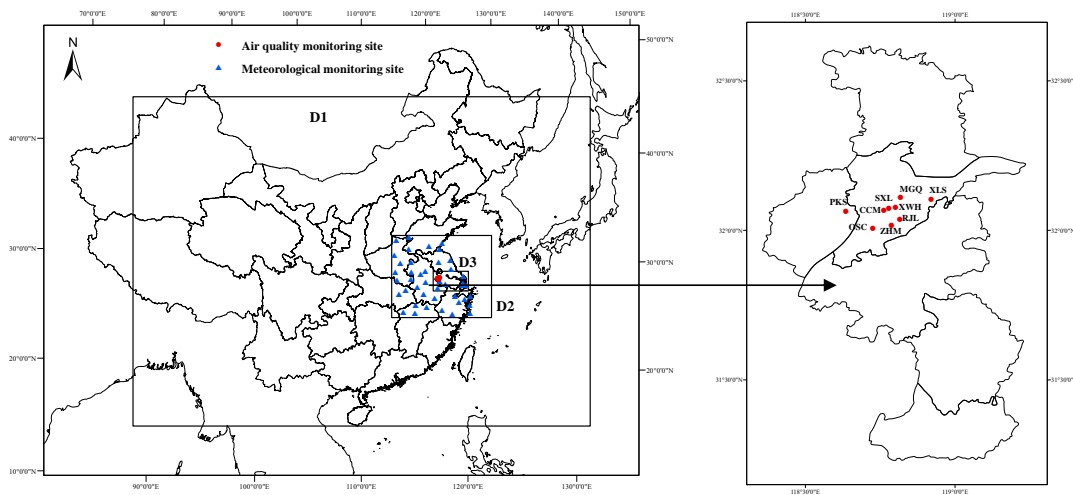





**Figure 2**

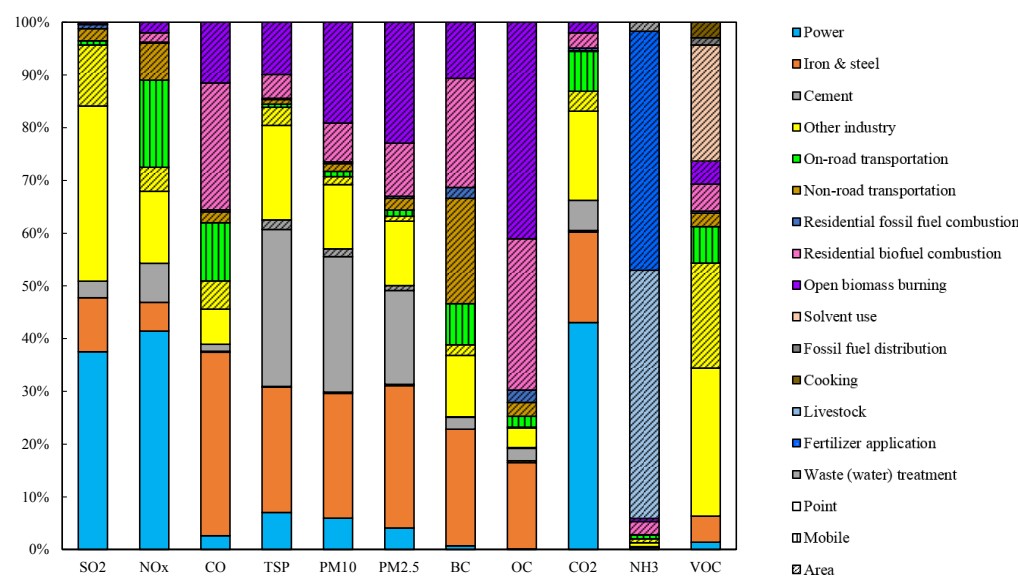



**Figure 3**

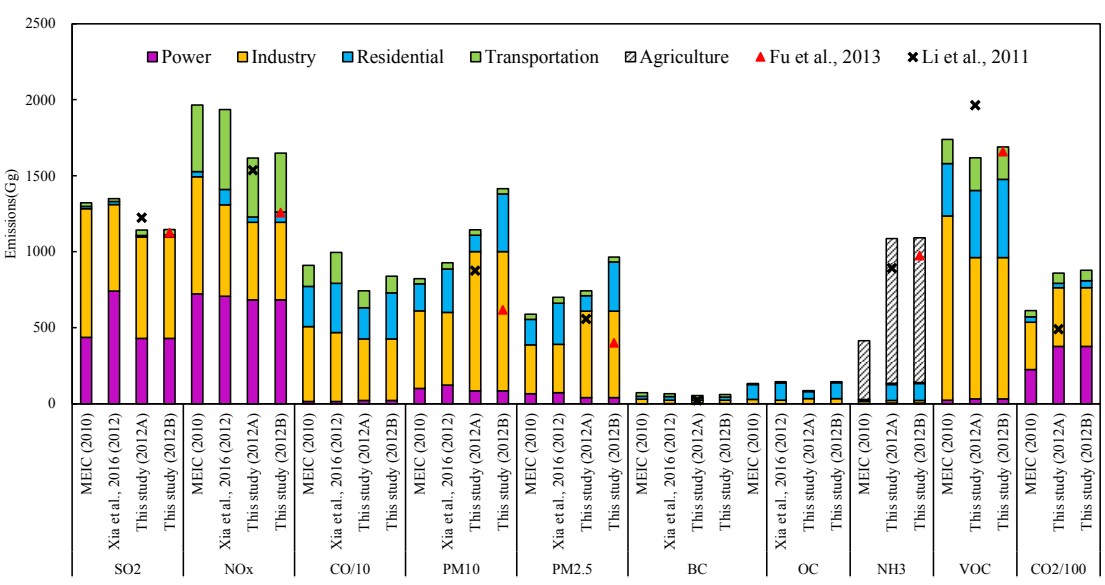





**Figure 4**

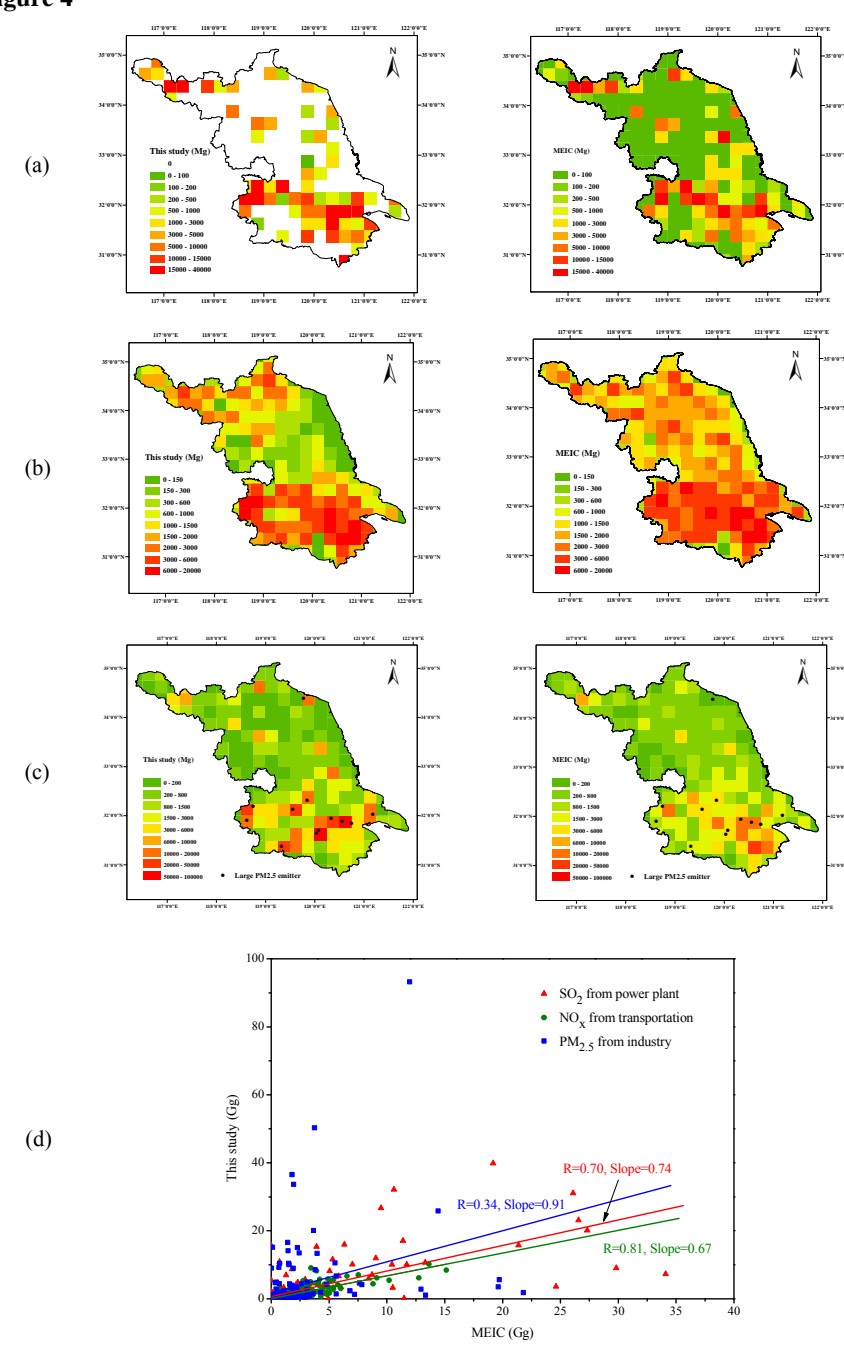





**Figure 5**

NO$_2$ VCDs   NOx emission   Linear regression

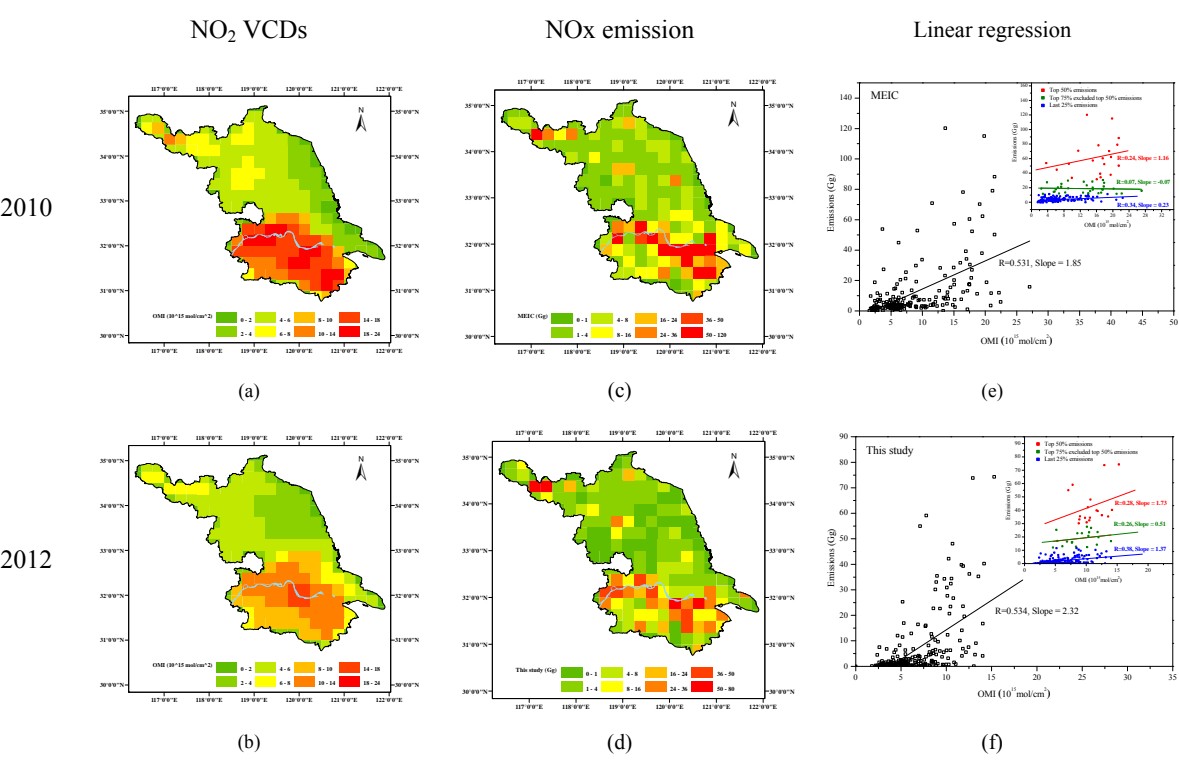

2010

2012

(a) (c) (e)

(b) (d) (f)





**Figure 6**

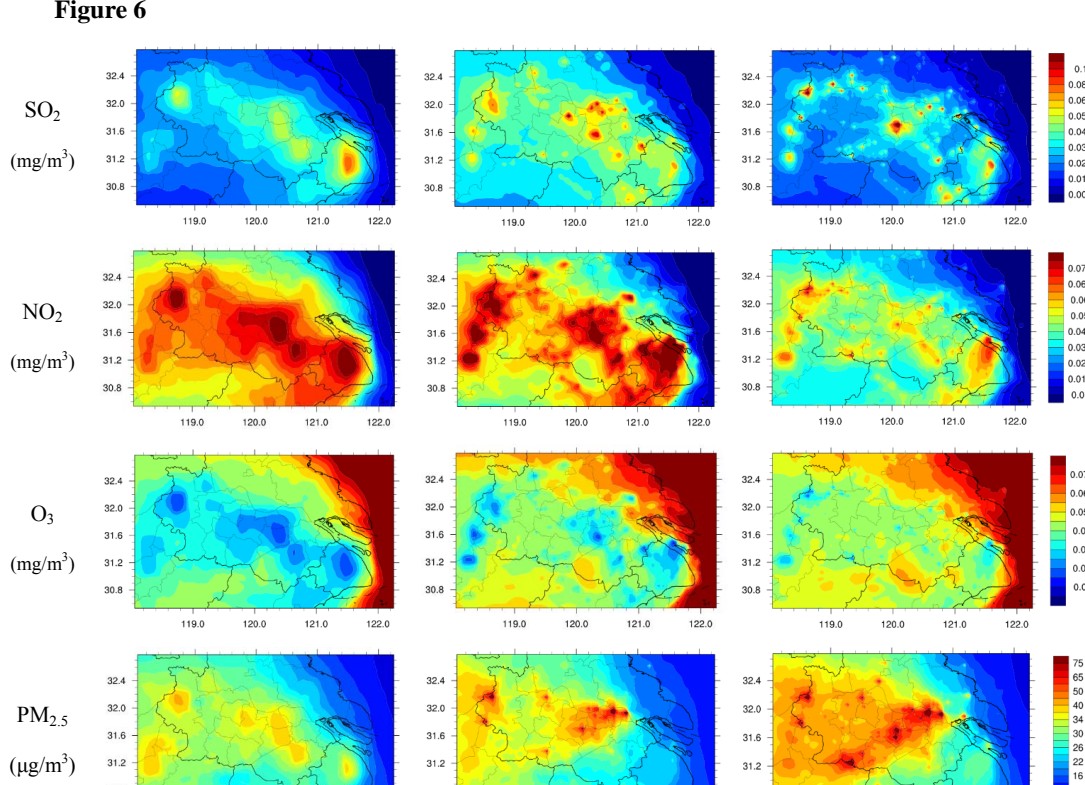

(a) National (MEIC)          (b) Regional (Fu et al., 2013)          (c) Provincial (this work)




**Figure 7**

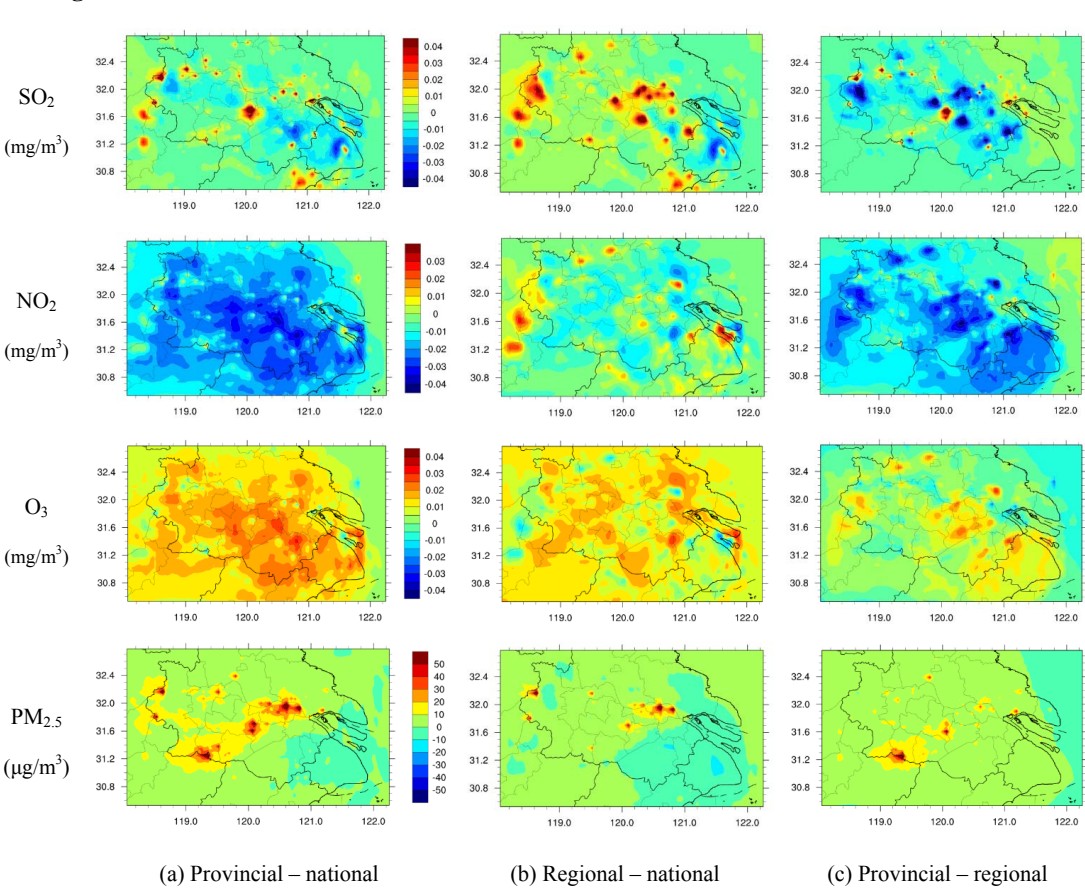

(a) Provincial – national          (b) Regional – national          (c) Provincial – regional





**Figure 8**

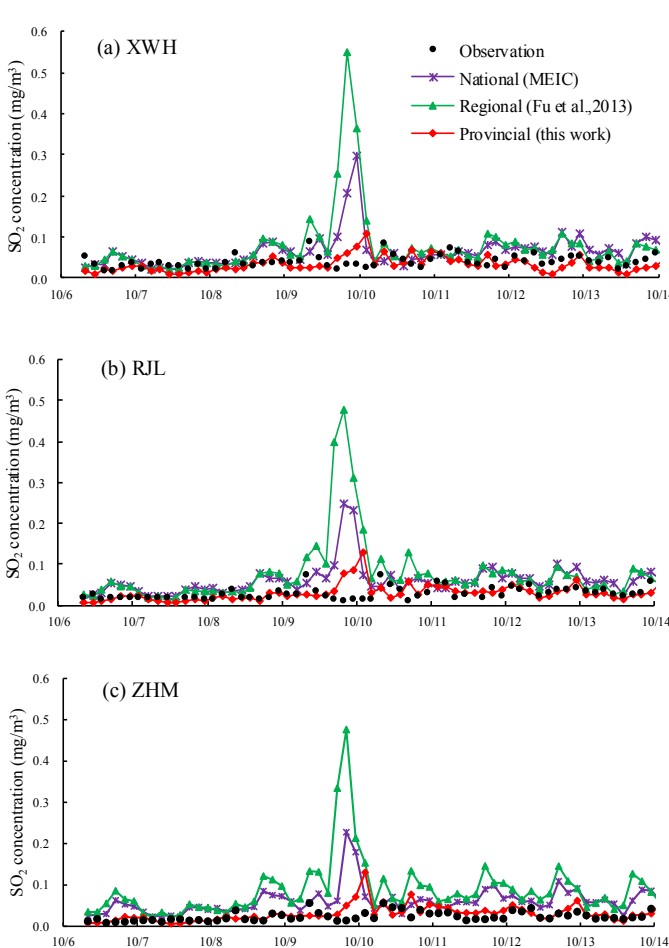





**Figure 9**

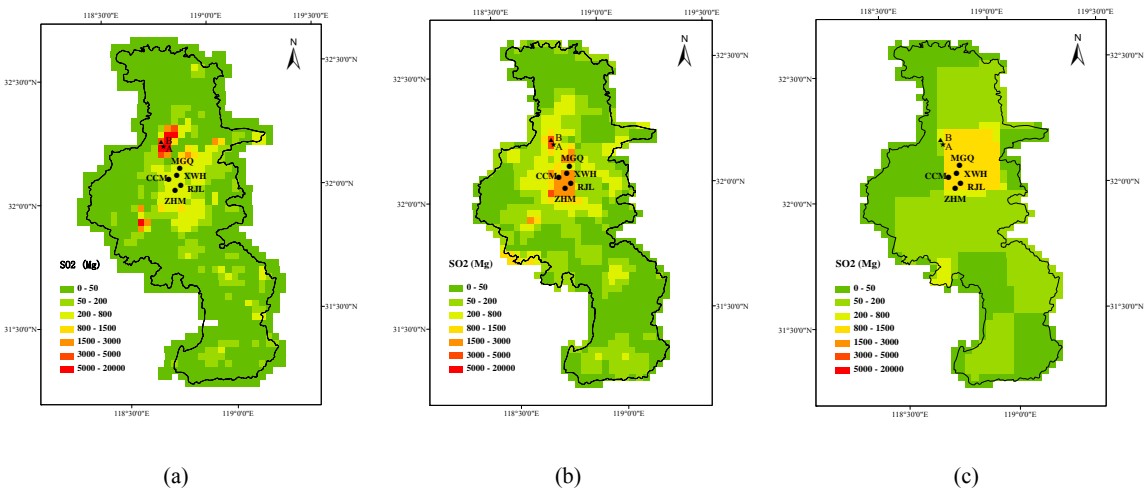

(a)  (b)  (c)





**Figure 10**

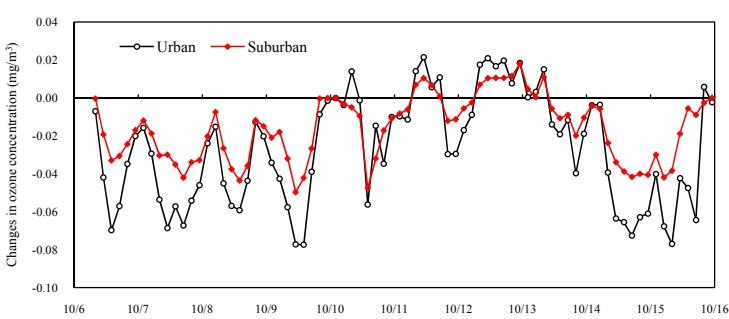

(a)

(b)