# Peer review of "Development of a high-resolution emission inventory and its evaluation through air quality modeling for Jiangsu Province, China"

_Atmospheric Chemistry and Physics, 2016_

## Referee Comment (RC1) · Anonymous Referee #1 · 3 Oct 2016

General comments: This paper presents the development of a provincial (Jiangsu) emission inventory, which can reduce the uncertainty in previous national level emission inventory and improve capability of air quality forecasting in Jiangsu province. The manuscript is well organized and written. Comprehensive evaluations were conducted, including comparisons with other inventories, with satellite data and with measurements of air pollutants. Another highlight of this manuscript is the inclusion of modeling results and sensitivity analysis of PM2.5 and O3 formation in Nanjing, which provides some implications of emission control strategies. The presentation of this newly developed high resolution emission inventory is suitable for being published in ACP. I only have some minor comments.

[Figure]
* * *
Specific comments:

Page 3, line 51-52: change "Issued in 2013, for example, the National Air Pollution Prevention Action Plan required" to "For example, the National Air Pollution Prevention Action Plan issued in 2013 required"

Page 9, line 232: The sentence "For other sources, the temporal distributions in Shanghai investigated by Li et al. (2011)." is not complete.

Page 11, line 284-285: D3 also includes some areas of Anhui, Zhejiang and Shanghai, but the provincial inventory was developed for Jiangsu. How were emissions in D3 treated for provincial inventory case? Please clarify.

Page 16, line 436-437: This implication might be questionable since the difference might be mainly driven by different methods (including different data sources) used in developments of emissions.

Sect. 3.3 shows the comparisons of pollutants from typical sources, how about pollutants from all sources? As shown in Figure 3, these typical sources account for significant but not entire amounts of the selected pollutants. The spatial distributions of total pollutants may substantially differ.

Page 19, line 514: NOx emissions in Jiangsu are only 65% higher than those of Shanghai. Given the large areas of Jiangsu, NOx emissions in south Jiangsu areas are comparable to those in Shanghai, and south Jiangsu is very close to Shanghai. It is hard to say local source in south Jiangsu is dominant. Is "those of Shanghai and Zhejiang Shanghai" a typo? Please correct it.

Sect. 4.2: Provincial emission inventory was developed but the innermost domain was only configured with focus on south Jiangsu. However, the newly developed emission in north Jiangsu might have impacts on the air quality simulations in south Jiangsu (as shown in Figure S6, some sources are from north). Besides, the evaluations using surface measurements were only shown in Nanjing, how about in other cities, particularly

in north Jiangsu?

Page 23, line 641-642: How were VOCs mapped to VOCs species? Were the same species profile applied for three emission inventories? Please clarify.

Figure S5: the wind vectors and colors are difficult to read, please make clearer plots.

Page 26, line 716-721: Please clarify the starting time in Figure S6. Are surface winds plotted in figure S5? Wind directions vary at different heights (for example, blue red and green lines show different directions), so it cannot imply the inconsistence with WRF results. The trajectories include vertical information, but the WRF winds are horizontal winds.

---

## Referee Comment (RC2) · Anonymous Referee #2 · 11 Oct 2016

General comments: This manuscript presents the development and evaluation of detailed emission inventories for major anthropogenic emission sources in Jiangsu Province, China. Emission is one of the major sources of uncertainty chemical transport modeling in China, where air pollution poses increasing threat to public health. Although it focuses only on one province (a relatively small emitter), which limits its scientific value to the broad ACP readership, this work could help establish procedures to develop and improve similar emission inventories in other provinces in this region. This manuscript could be improved in several key aspects to meet the requirements ACP. First, the presentation, in the current form, needs to be polished. There are several places in the text that the expression is either ambiguous or confusing. Second,

the manuscript lacks detailed information on how these inventories were compiled (formula, parameters, etc), and where to find the key information to verify or reproduce the results reported here. Third, some of the materials included here are not relevant to the main theme. For example, the sensitivity of ozone and PM2.5 to 50% reduction of NOx and VOC emissions has nothing to do with development and evaluation of the emission inventories. The authors may consider either broadening the theme (development, evaluation, and application of the EIs) or removing the sensitivity section. Finally, the interpretation of some results need to be toned down or adjusted. Detailed comments and suggested changes are provided below.

Specific comments:

L6: process(es).

L26-30: Please reword this sentence. Define "unfavorable meteorology". Is the meteorology simulated by WRF inaccurate or the meteorology condition not conducive to pollution formation and accumulation? If the meteorology simulation is problematic, emission may reduce the model bias by offsetting opposite bias caused by meteorology inputs, but that does not warrant the quality of emission data per se.

Section 2.1: There are at least eight sectors considered here. Is Eq. (1) applied to all sectors or only to point sources? In the case that other formulas are used, please explain in more details (only citations are briefly provided here). It will be very useful to provide a table that includes the following key information: 1) sector; 2) sources included in that sector; 3) method to estimate emission; 4) key parameters and data sources to obtain the information; 5) special adjustment made.

L177: which model requirement?

L181-185: Can not understand this sentence. Please rephrase it. It seems a subtraction is involved here, by using data from different sources. What is the implication for uncertainty by subtracting data from different sources?

Section 2.3. Again, a list of emission factors for each sector/source will be very useful. Maybe a table in the supplementary information.

L227: "by with"? Please revise the sentence.

L233-240. Please provide information of the spatial maps used to distribute emissions for each sector. Currently only the data for open burning have been given.

L280: "initial (concentration) and boundary conditions"

L290-291: Please elaborate how the vertical distribution is determined here. Is this applied to all sectors or just point sources?

L447-449: Is the agricultural GDP increase due to change in market price or commodity quantity?

Section 2 may need a sub-section to describe the measurement data (satellite and ground observations) used in this study.

Section 4.1 Using satellite NO2 data for emission evaluation has been well explored (check the literature for more details). There are several issues related to the method used here. Note OMI observes VCD, not emission. Therefore it may make more sense to compare the VCD from OMI to the equivalent from CMAQ, not the emission distribution directly, unless the nonlinear relationship between emission and VCD is accounted for. In addition the noise to signal ratio in OMI NO2 increases with decreasing VCD. The interpretation of the correlation between emission and OMI VCD needs to consider these factors.

L549-550: See the comments above.

Section 4.3: It is good to see that the new emission inventories can reduce high bias during extreme events when the meteorology is not correctly simulated. Do we have any results showing that this emission data can be used to improve prediction of real pollution events? This manuscript discusses the new emission data for the

entire province. Why is only the model prediction over Nanjing discussed in the model-observation comparison?

Section 5. As mentioned earlier, this section may not be necessary to support the main argument here. The interpretation of the results needs to acknowledge several caveats in the design and scope of these simulations. For instance, the brute force method does not consider nonlinearity in ozone response to precursor change. Large uncertainties exist in the emission dataset, including VOCs that have not been evaluated here, which will affect the chemistry and directionality of the ozone response. In addition, the sensitivity examines ozone response, but NOx and VOCs changes also result in changes in other pollutants, such as PM2.5. It is hence premature to draw such conclusion as in L770-772.

---

## Author Comment (AC1) · 22 Nov 2016

Manuscript No.: acp-2016-567

Title: Development of a high-resolution emission inventory and its evaluation and application through air quality modeling for Jiangsu Province, China

Authors: Yaduan Zhou, Yu Zhao, Pan Mao, Qiang Zhang, Jie Zhang, Qiu Liping, Yang Yang

We thank very much for the valuable comments and suggestions from reviewer 1, which help us improve our manuscript. The comments were carefully considered and revisions have been made in response to suggestions. Following is our point-by-point

responses to the comments and corresponding revisions.

Reviewer #1

1. General comments: This paper presents the development of a provincial (Jiangsu) emission inventory, which can reduce the uncertainty in previous national level emission inventory and improve capability of air quality forecasting in Jiangsu province. The manuscript is well organized and written. Comprehensive evaluations were conducted, including comparisons with other inventories, with satellite data and with measurements of air pollutants. Another highlight of this manuscript is the inclusion of modeling results and sensitivity analysis of PM2.5 and O3 formation in Nanjing, which provides some implications of emission control strategies. The presentation of this newly developed high resolution emission inventory is suitable for being published in ACP.

Response and revisions:

We appreciate the reviewer's positive remarks on our manuscript.

2. Line 51-52: change "Issued in 2013, for example, the National Air Pollution Prevention Action Plan required" to "For example, the National Air Pollution Prevention Action Plan issued in 2013 required".

Response and revisions:

We thank the reviewer's suggestion and the sentence has been corrected as suggested in lines 51-52 Page 3 in the revised manuscript.

3. line 232: The sentence "For other sources, the temporal distributions in Shanghai investigated by Li et al. (2011)." is not complete.

Response and revisions:

We thank the reviewer's reminder. The sentence has been revised as "For other sources, the temporal distributions for Shanghai investigated by Li et al. (2011) were adopted" in lines 241-242 Page 10 in the revised manuscript.

4. line 284-285: D3 also includes some areas of Anhui, Zhejiang and Shanghai, but the provincial inventory was developed for Jiangsu. How were emissions in D3 treated for provincial inventory case? Please clarify.

Response and revisions:

We thank the reviewer's reminder and admit that we did not clarify the emission inventory applied in the provincial modeling case. In the provincial case, the emissions for the regions outside Jiangsu in D3 were obtained from the relocated 4×4 km regional inventory developed by Fu et al. (2013). Corresponding revision was shown in lines 591-593 Page 22 in the revised manuscript.

5. line 436-437: This implication might be questionable since the difference might be mainly driven by different methods (including different data sources) used in developments of emissions.

Response and revisions:

We thank the reviewer's comment and agree that the implication could not be sufficiently supported. The sentence has been removed in the revised manuscript.

6. Sect. 3.3 shows the comparisons of pollutants from typical sources, how about pollutants from all sources? As shown in Figure 3, these typical sources account for significant but not entire amounts of the selected pollutants. The spatial distributions of total pollutants may substantially differ.

Response and revisions:

We thank the reviewer's comment. The main purpose of Section 3.3 is not to compare the spatial distribution of the total emissions between inventories, but to investigate the influence of different methods and data sources on the spatial distribution of pollutants in emission estimation. Therefore emissions from selected sources are included in the comparisons between inventories at national and provincial scales. For example, industrial combustion was generally treated as area sources in the national inventory,

while information of 6,194 point sources was collected and compiled in this work to develop the provincial inventory. The linear regression analysis between gridded emissions of industrial combustion in provincial and national inventories thus implies the deviations in spatial distributions of emissions resulting from different methods and data.

7. Page 19, line 514: NOx emissions in Jiangsu are only 65% higher than those of Shanghai. Given the large areas of Jiangsu, NOx emissions in south Jiangsu areas are comparable to those in Shanghai, and south Jiangsu is very close to Shanghai. It is hard to say local source in south Jiangsu is dominant. Is "those of Shanghai and Zhejiang Shanghai" a typo? Please correct it.

Response and revisions:

We thank the reviewer's reminder and admit a mistake in the original sentence. Jiangsu's NOX emissions were estimated 282% instead of 65% higher than Shanghai's in MEIC, and they were also 65% and 94% larger than those of another two contiguous provinces Zhejiang and Anhui, respectively. According to the YRD regional inventory by Fu et al. (2013), moreover, NOx emissions of Suzhou and Wuxi in southern Jiangsu were nearly twice of those for the nearby cities Jiaxing and Huzhou in northern Zhejiang. Therefore we believe the local emissions played an important role in air pollution level in southern Jiangsu. In lines 537-538 Page 20 in the revised manuscript, the word "dominated" has been deleted and the sentence has been modified accordingly.

8. Sect. 4.2: Provincial emission inventory was developed but the innermost domain was only configured with focus on south Jiangsu. However, the newly developed emission in north Jiangsu might have impacts on the air quality simulations in south Jiangsu (as shown in Figure S6, some sources are from north). Besides, the evaluations using surface measurements were only shown in Nanjing, how about in other cities, particularly in north Jiangsu?

Response and revisions:

We thank the reviewer's comment. Compared to northern regions, intensive economy and industry are located in the more developed southern Jiangsu, thus the air quality issue is more concerned in the region. In this work, the emissions of $SO_2$, $NO_x$, CO, TSP, $PM10$, $PM2.5$, BC, OC, $CO_2$ and VOCs in the five cities in southern Jiangsu were estimated to contribute collectively 54.1%, 53.3%, 47.2%, 52.4%, 48.8%, 48.6%, 42.7%, 30.2%, 56.3% and 48.4% to the total emissions of the whole province in 2012, respectively. As shown in Figure S3, grids with relative large emissions (red color) were mainly clustered in southern Jiangsu. Therefore, we focused mainly on southern Jiangsu in the air quality simulation. As the emissions from northern Jiangsu were just applied in D2 simulation that provided the boundary conditions for D3 simulation, we believe the effects of those emissions were limited.

Regarding the model evaluation, the official data on air quality from surface measurements have been routinely published since 2013, thus the complete data for Jiangsu were unavailable for the years before 2013, except for its capital city Nanjing. Therefore only the observations at the nine sites in Nanjing were applied in this work.

9. Page 23, line 641-642: How were VOCs mapped to VOCs species? Were the same species profile applied for three emission inventories? Please clarify.

Response and revisions:

We thank the reviewer's comment. In this work, total NMVOC emissions for given source type in Table S1 were broken down into individual species using Eq. (1):

$$E(i,k)=E(i)\times X(i,k) \quad (1)$$

where E and X are the emissions and the chemical profile of VOCs (%), respectively; i and k represent the source type and individual VOCs species, respectively.

Different species profiles were applied in various inventories. The chemical profiles were mainly taken from domestic measurements, including residential fossil fuel and biomass burning, open biomass burning, on-road transportation, iron & steel, paint

production, solvent use and oil refineries. For sources without sufficient local measurements, results from foreign studies were applied including the SPECIATE database by USEPA (2014). To reduce the possible uncertainty of source profile from individual measurement, Li et al. (2014) developed the "composite profiles" for sources where multiple candidate profiles were available, by revising the oxygenated volatile organic compounds (OVOCs) fraction and averaging the fractions in different profiles for each species. In this work, "composite profiles" were updated following the method by Li et al. (2014), and the most recent source profiles from domestic results were contained.

To meet the requirement of CMAQ modeling, VOCs emissions were assigned to chemical mechanism (Carbon Bond 05) species by multiplying the emissions of individual species and mechanism-specific conversion factors:

$$E(i,m)=E(i,k) \times C(k,m)/M(k) \quad (2)$$

where E, M, and C are the emissions, mole weight, and the conversion factor, respectively; i, m, and k represent the source type, individual species, and the chemical mechanism species, respectively.

All the details could be found in another paper specifically for the development and evaluation of provincial VOC emission inventory (Zhao et al., in preparation), and we have stated that in lines 232-234 Page 9, and in lines 681-683 Page 25 in the revised manuscript.

10. Figure S5: the wind vectors and colors are difficult to read, please make clearer plots.

Response and revisions:

We thank the reviewer's suggestion and an updated Figure S5 with clearer plots has been provided in the revised supplement.

11. Page 26, line 716-721: Please clarify the starting time in Figure S6. Are surface winds plotted in figure S5? Wind directions vary at different heights (for example, blue

red and green lines show different directions), so it cannot imply the inconsistence with WRF results. The trajectories include vertical information, but the WRF winds are horizontal winds.

Response and revisions:

We thank the reviewer's reminder. HYSPLIT 24 h back-trajectories at 50, 250 and 500 m are calculated every 6 hours starting at 11pm on 9th October and ending at 5 am on 8th October (The caption of Figure S6 in the supplement has been revised). Figure S6 showed the air flow at the height of 50, 100 and 200m originated from Xuanwuhu site. Given the height of the first modeling layer in WRF was set at 50 m, the back trajectory at 50 m should be more suitable to indicate the air transport from 5 am on 8th to 11 pm on 9th October. As indicated in Figure S6 in the supplement, the air mass at 50 m altitude (the red line) came from northeast at 11pm on 9th October. However, the result was notably different with that simulated by WRF, in which the dominant wind direction was northwest ($150°$-$170°$) at that time.

References Fu, X., Wang, S. X., Zhao, B., Xing, J., Cheng, Z., Liu, H., and Hao, J. M.: Emission inventory of primary pollutants and chemical speciation in 2010 for the Yangtze River Delta region, China, Atmos. Environ., 70, 39-50, 2013.

Li, M., Zhang, Q., Streets, D. G., He, K. B., Cheng, Y. F., Emmons, L. K., Huo, H., Kang, S. C., Lu, Z., Shao, M., Su, H., Yu, X., and Zhang, Y.: Mapping Asian anthropogenic emissions of non-methane volatile organic compounds to multiple chemical mechanisms, Atmos. Chem. Phys., 14, 5617-5638, 2014.

U.S. Environmental Protection Agency (USEPA): SPECIATE Version 4.4, available at: https://www3.epa.gov/ttnchie1/software/speciate (last access: 12 November 2015), 2014.

Zhao, Y., Mao, P., Zhou, Y., Yang, Y., Zhang, J., Wang, S., Dong, Y., Xie, F., Yu, Y., and Li, W.: Improved provincial emission inventory and speciation profiles of anthropogenic non-methane volatile organic compounds: a case study for Jiangsu, China, in preparation.

---

## Author Comment (AC2) · 22 Nov 2016

Manuscript No.: acp-2016-567

Title: Development of a high-resolution emission inventory and its evaluation and application through air quality modeling for Jiangsu Province, China

Authors: Yaduan Zhou, Yu Zhao, Pan Mao, Qiang Zhang, Jie Zhang, Qiu Liping, Yang Yang

We thank very much for the valuable comments and suggestions from reviewer 2, which help us improve our manuscript. The comments were carefully considered and revisions have been made in response to suggestions. Following is our point-by-point

responses to the comments and corresponding revisions.

Reviewer #2

1. This manuscript presents the development and evaluation of detailed emission inventories for major anthropogenic emission sources in Jiangsu Province, China. Emission is one of the major sources of uncertainty chemical transport modeling in China, where air pollution poses increasing threat to public health. Although it focuses only on one province (a relatively small emitter), which limits its scientific value to the broad ACP readership, this work could help establish procedures to develop and improve similar emission inventories in other provinces in this region. This manuscript could be improved in several key aspects to meet the requirements ACP. First, the presentation, in the current form, needs to be polished. There are several places in the text that the expression is either ambiguous or confusing. Second, the manuscript lacks detailed information on how these inventories were compiled (formula, parameters, etc), and where to find the key information to verify or reproduce the results reported here. Third, some of the materials included here are not relevant to the main theme. For example, the sensitivity of ozone and PM2.5 to 50% reduction of NOx and VOC emissions has nothing to do with development and evaluation of the emission inventories. The authors may consider either broadening the theme (development, evaluation, and application of the EIs) or removing the sensitivity section. Finally, the interpretation of some results need to be toned down or adjusted.

Response and revisions:

We appreciate the reviewer's crucial and important comments. In general, the presentation of the work has been improved, based on specific comments/suggestion from the reviewer. Detailed information on emission inventory development has been added, as indicated in the response to Questions 4, 7, 9, and 11 from the reviewer. In particular, Table S1 has been expanded and a new Table S3 has been added in the supplement to provide more detailed information on data sources of activities and emission factors.

We also take the reviewer's suggestion and broaden the theme by adding the word "application" in the title of the paper. Uncertainties of the analysis have been also discussed to avoid overstatement of the work, as indicated in the response to Questions 14 and 16 from the reviewer.

2. L6: process(es).

Response and revisions:

We thank the reviewer's reminder and it is now corrected.

3. L26-30: Please reword this sentence. Define "unfavorable meteorology". Is the meteorology simulated by WRF inaccurate or the meteorology condition not conducive to pollution formation and accumulation? If the meteorology simulation is problematic, emission may reduce the model bias by offsetting opposite bias caused by meteorology inputs, but that does not warrant the quality of emission data per se.

Response and revisions:

The term "unfavorable meteorology" mentioned in the mansucript referred to the meteorology condition with low wind speed and PBL height, in which horizontal and vertical movement of atmosphere was limited. Under such condition, primary pollutants from large emitters would be easily accumulated over time, and air quality would thus be largely influenced by local sources. The sentence has been revised as "Under the unfavorable meteorology in which horizontal and vertical movement of atmosphere was limited, the simulated $SO_2$ concentrations at downtown Nanjing (the capital city of Jiangsu) using the regional or national inventories were much higher than observation, implying the overestimated urban emissions when economy or population densities were applied to downscale or allocate the emissions" in lines 26-31 Pages 2-3 in the revised manuscript.

We also agree with the reviewer that uncertainties existed in meteorology field simulation and that the performance of air quality modeling might be offset by the opposite

bias of emission inventory and simulated meteorology. To test the deviation caused by meteorology field simulation with WRF, the model performance of SO2 was reevaluated excluding the data from 5pm Oct 9th to 5am 10th. The normalized mean bias (NMB) were recalculated at -13%, 66%, and 50%, and the normalized mean errors (NME) were 45%, 88%, and 78% using the provincial, regional, and national inventories, respectively. This result thus implied that the provincial inventory could better support the air quality modeling, even uncertainties existed in meteorology field simulation.

4. Section 2.1: There are at least eight sectors considered here. Is Eq. (1) applied to all sectors or only to point sources? In the case that other formulas are used, please explain in more details (only citations are briefly provided here). It will be very useful to provide a table that includes the following key information: 1) sector; 2) sources included in that sector; 3) method to estimate emission; 4) key parameters and data sources to obtain the information; 5) special adjustment made.

Response and revisions:

We thank the reviewer's important comment. We checked the sectors, and confirmed that there are seven main sectors (residential & commercial is one sector). We have stated in line 133 Page 6 in the revised manuscript that Eq. (1) is applied only to point sources. No equation is provided for mobile sources (i.e., on-road vehicles) as their emissions were estimated using COPERT model. We have also provided the equation and explanation for calculating the emissions from area sources in lines 148-151 Page 7 in the revised manuscript. As suggested by the reviewer, the detailed source categories by sector and the main data sources of activity levels by category were summarized in the revised Table S1 in the supplement. We keep the discussions on data sources of emission factors by category in Section 2.3, with the detailed information provided in Table S2 and a new Table S3 in the supplement, as explained in the response to Q7.

5. L177: which model requirement?

Response and revisions:

We thank the reviewer's reminder and admit the original word was confusing. The "model" here indicates COPERT 4 applied for calculating the emissions from on-road vehicles. Since differences exist in the categories of vehicle types between Chinese and European criteria, population of each Chinese vehicle category should be converted to the number consistent with the COPERT categories, in order to use the model. According to Cai and Xie (2007), for example, the buses, passenger cars (PC), heavy-duty trucks (HDT), light-duty trucks (LDT), and motorcycles (MT) in COPERT 4 indicate the big-size and middle-size passenger cars, small-size and mini passenger cars, heavy-duty and intermediate duty trucks, light-duty and mini trucks, and motorcycles in Chinese statistic system. The original sentence has been modified as "Populations of different vehicle types were derived from statistical yearbooks by city and then converted to the numbers in COPERT 4 categories" in lines 183-184 Page 8 in the revised manuscript.

6. L181-185: Can not understand this sentence. Please rephrase it. It seems a subtraction is involved here, by using data from different sources. What is the implication for uncertainty by subtracting data from different sources?

Response and revisions:

We thank the reviewer's important comment. In this work, as indicated in lines in the revised manuscript, information from the official environmental statistics, Pollution Source Census (PSC), and on-site survey on large emitters were collected to estimate the emissions from point sources. Although most of large industry sources could be investigated through this method, information of certain small plants were not included in those datasets and had to be treated as area sources. Thus the energy consumption and production provided in provincial statistical yearbooks were applied to check the completeness of point source investigation, and the difference between the energy statistics and overall activity levels of point sources was assumed as the activity levels of industrial area sources. The sentence has been revised as "For area sources, the coal consumption of residential activities was directly taken from National Energy Statistic Yearbook (NBSC, 2013c), while that of small industrial plants were calculated by subtracting the coal consumed by industrial point sources from the coal consumption of total industry provided in the provincial energy balance (NBSC, 2013c)" in lines 188-192 Page 8 in the revised manuscript.

7. Section 2.3. Again, a list of emission factors for each sector/source will be very useful. Maybe a table in the supplementary information.

Response and revisions:

We thank the reviewer's important comment. As required by the reviewer, a new Table S3 was provided in the revised supplement, summarizing the detailed emission factors and their sources by sector.

8. L227: "by with"? Please revise the sentence.

Response and revisions:

We thank the reviewer's reminder and admit the typo error. Now the error has been corrected in line 237 Page 10 in the revised manuscript.

9. L233-240. Please provide information of the spatial maps used to distribute emissions for each sector. Currently only the data for open burning have been given.

Response and revisions:

We thank the reviewer's suggestion and agree that more detailed information on spatial distribution of emissions should be provided. For point sources, information of latitude and longitude for each plant was collected from PSC. Locations of the total 6750 point sources shown in Figure S2 in the supplement were verified and modified in Google Earth. The road and hydrographic net in Jiangsu were respetively used to distribute the emissions from on-road transportation and ships city-by city. Densities

of GDP and population in Jiangsu applied to allocate emissions of area sources were from the research by Huang et al. (2014) and Fu et al. (2014). NH3 emissions from livestock, fertilizer usage and agricultural vehicles were allocated by incorporating the population density and distribution of land types categorized to agricultural activities from the land cover dataset GlobCover2009 (http://globalchange.nsdc.cn). Such information has been added in lines 243-256 Page 10 in the revised manuscript.

10. L280: "initial (concentration) and boundary conditions"

Response and revisions:

We thank the reviewer's reminder and the text has been corrected in line 293 Page 11 in the revised manuscript.

11. L290-291: Please elaborate how the vertical distribution is determined here. Is this applied to all sectors or just point sources?

Response and revisions:

We thank the reviewer's important suggestion and admit that it was not clearly explained in the original manuscripts. The vertical distributions of emissions were directly taken from L. Wang et al. (2010) except for the power sector, as the height of discharge outlet for each power plant was available for Jiangsu. According to L. Wang et al. (2010), the fractions of emissions of industry sources were 50%, 30% and 20% in layers 1-3, respectively. For the sources near the surface, i.e., transportation, residential & commercial combustion, solvent use, agriculture, and other sources, emissions were overall allocated to the first vertical layer in the model. The emissions of power plants were concentrated in layers 2-5, with the fractions estimated as 14.7%, 45.7%, 34.9% and 4.7%, respectively, based on the height information of the stacks. We have added the information in lines 304-311 Page 12 in the revised manuscript.

12. L447-449: Is the agricultural GDP increase due to change in market price or commodity quantity?

Response and revisions:

We thank the reviewer's crucial comment and agree that application of commodity quantity would be more reasonable and persuasive to illustrate the case. We have checked the growth of livestock and poultry in Jiangsu from the provincial statistics. In lines 465-468 Page 17 in the revised manuscript, the sentence has been modified as "According to the provincial statistics, the total numbers of livestock and poultry increased 6% and 10% from 2010 to 2012 in Jiangsu (JSNBS, 2013). The growth of activity levels was expected to result in enhanced emissions, as very little progress was achieved for NH3 control for these years."

13. Section 2 may need a sub-section to describe the measurement data (satellite and ground observations) used in this study.

Response and revisions:

We thank the reviewer's comment. The ground observation and retrieval of satellite data were not the main tasks of this study, and the data were directly taken from other publication/groups without any extra modification. Therefore, we do not think it is necessary to describe the data in a specific section, and we have kept the description in corresponding sections in the revised manuscript, i.e., lines 593-599 for ground observations and lines 526-534 Pages 19-20 for satellite observation.

14. Section 4.1 Using satellite NO2 data for emission evaluation has been well explored (check the literature for more details). There are several issues related to the method used here. Note OMI observes VCD, not emission. Therefore, it may make more sense to compare the VCD from OMI to the equivalent from CMAQ, not the emission distribution directly, unless the nonlinear relationship between emission and VCD is accounted for. In addition, the noise to signal ratio in OMI NO2 increases with decreasing VCD. The interpretation of the correlation between emission and OMI VCD needs to consider these factors.

[Figure]

Response and revisions:

We thank the reviewer's very important comment and agree that the uncertainty of the method should be discussed in the paper. First, the quality control and reliability of OMI retrieved NO2 VCDs has been added in lines 529-534 Pages 19-20 in the revised manuscript with literature provided. We then acknowledged that the comparisons between spatial distribution of NO2 VCDs and emissions should be cautiously interpreted particularly for regions with relatively low values, as the noise to signal ratio in OMI NO2 increases with decreased VCDs. Although summer NO2 VCDs were applied in this study to eliminate the effects of long lifetime of NO2 on pollution plums transport and chemical reaction, non-linear relationship still exists between VCDs and emissions. More detailed comparisons between NO2 from satellite observation and CTM are thus recommended when improved characterization of NO2 vertical distribution is available for the region. We have added such discussions in lines 576-582 Page 21 in the revised manuscript.

15. Section 4.3: It is good to see that the new emission inventories can reduce high bias during extreme events when the meteorology is not correctly simulated. Do we have any results showing that this emission data can be used to improve prediction of real pollution events? This manuscript discusses the new emission data for the entire province. Why is only the model prediction over Nanjing discussed in the model-observation comparison?

Response and revisions:

We thank the reviewer's positive remarks and suggestion. As shown in Table 2, better model performances for SO2, NOx, O3 and PM2.5 simulation could be achieved with provincial inventory than those with national or regional inventory for the whole October, implying the improvement of provincial emission inventory for Jiangsu. Besides the special case discussed in Section 4.3, the simulated PM2.5 concentrations from 8pm on 18th to 5pm on 19th October with different inventories could also indicate the

advantage of the provincial inventory against the regional and national inventories in a pollution event prediction. Taking Caochangmen (CCM) site as the example, the observed PM2.5 concentration kept increasing from 8pm on 18th with the highest value reaching 114 $\mu$g/m3 at 2am on 19th. Simulated PM2.5 concentrations with provincial, regional and national inventory at that time were 90, 53 and 45 $\mu$g/m3, respectively. The correlation coefficients between observations and simulations with the three inventories were calculated as 0.66, 0.44 and 0.30 in the episode, respectively, indicating the better performance with provincial inventory in real pollution episode simulation. We have added such discussions in lines 619-626 Page 23 in the revised manuscript.

Regarding the second question, the official data on air quality from surface measurements have been routinely published since 2013, thus the complete data for Jiangsu were unavailable for the years before 2013, except for its capital city Nanjing. Therefore only the observations at the nine sites in Nanjing were applied in this work.

16. As mentioned earlier, this section may not be necessary to support the main argument here. The interpretation of the results needs to acknowledge several caveats in the design and scope of these simulations. For instance, the brute force method does not consider nonlinearity in ozone response to precursor change. Large uncertainties exist in the emission dataset, including VOCs that have not been evaluated here, which will affect the chemistry and directionality of the ozone response. In addition, the sensitivity examines ozone response, but NOx and VOCs changes also result in changes in other pollutants, such as PM2.5. It is hence premature to draw such conclusion as in L770-772.

Response and revisions:

We thank the reviewer's very crucial comment and suggestion. As suggested, we revised the title of the paper, stressing the "application" of the emission inventory. We agree with the reviewer that the brute-force method ignores the nonlinearity of O3 response to the changes of precursor emissions, and that is a big uncertainty of the anal-

ysis. The results should thus be cautiously interpreted, and the comparisons with other simulation methods that take the nonlinearity mechanisms into account (e.g., OSAT or tagged species method) are further recommended. We have added the discussions in lines 825-831 Page 30 in the revised manuscript.

For VOC, the detailed information on development and evaluation of provincial inventory with source profiles is presented in a separate paper (Zhao et al., in preparation). Regarding the length of current paper, we have briefly stated that in lines 232-234 Page 9, and in lines 681-683 Page 25 in the revised manuscript.

We also agree with the reviewer that the air quality includes many issues besides O3 problem, thus the sentence in lines 770-772 in the original manuscript has been deleted to avoid overstating the case.

References

Cai, H., Xie, S. D.: Estimation of vehicular emission inventories in China from 1980 to 2005, Atmos. Environ., 41, 8963-8979, 2007.

Fu, J. Y., Jiang, D., Huang, Y. H.: 1 Km Grid Population Dataset of China (PopulationGrid_China), Global Change Research Data Publishing & Repository, DOI: 10.3974/geodb.2014.01.06. V1, 2014.

Huang, Y. H., Jiang, D., Fu, J. Y.: 1 Km Grid GDP Data of China (2005, 2010) (GDPGrid_China), Global Change Research Data Publishing & Repository, DOI: 10.3974/geodb.2014.01.07. V1, 2014.

JSNBS (Jiangsu Bureau of Statistics): Statistical Yearbook of Jiangsu, Beijing, China Statistics Press, 2013 (in Chinese).

Wang, L. T., Jang, C., Zhang, Y., Wang, K., Zhang, Q., Streets, D. G., Fu, C. J., Lei, Y., Schreifels, J., He, K. B., Hao, J. M., Lam, Y. F., Lin, J., Meskhidze, N., Voorhees S., Evarts D., Phillips S.: Assessment of air quality benefits from national air pollution control policies in China. Part II: Evaluation of air quality predictions and air quality

benefits assessment. Atmos. Environ., 44, 3449-3457, 2010.

Zhao, Y., Mao, P., Zhou, Y., Yang, Y., Zhang, J., Wang, S., Dong, Y., Xie, F., Yu, Y., and Li, W.: Improved provincial emission inventory and speciation profiles of anthropogenic non-methane volatile organic compounds: a case study for Jiangsu, China, in preparation.

————————————————